# STC3D: Self-Supervised Contrastive Learning with Spatial Transformations for 3D Medical Image Analysis

## Abstract

Self-Supervised Learning (SSL) has demonstrated promising results in 3D medical image analysis, but traditional SSL methods often lack high-level semantics during pre-training, limiting performance on downstream tasks. Recent methods like Volume Contrast (VoCo) have addressed this by leveraging contextual position priors in 3D images, but VoCo relies on random cropping, which may reduce robustness to anatomical variations. In this paper, we propose STC3D, a novel SSL framework that applies controlled spatial transformations (rotation, translation, scaling) to generate multiple views of 3D volume images. These transformed views are then used for contrastive learning, enhancing invariance to anatomical structure transformations. Additionally, STC3D includes a regularization branch to promote feature discrepancy between different base slices, improving the discriminative power of learned representations. Experimental results on several benchmark datasets, including BTCV, MSD Spleen, MM-WHS, and BraTS 21, demonstrate that STC3D outperforms existing methods in segmentation and classification tasks.

## 1 Introduction

Deep learning has achieved remarkable progress in 3D medical image analysis, automating complex tasks such as segmentation, classification, and detection. Despite these advancements, the adoption of deep learning is hindered by the high cost of obtaining expert annotations, which are time-consuming and require domain expertise. As a result, the lack of labeled data remains a significant challenge for the development and deployment of deep learning models in medical image analysis.

To address this challenge, Self-Supervised Learning (SSL) has emerged as a promising alternative, allowing models to learn from unlabeled data while maintaining strong performance. SSL leverages pretext tasks to teach models useful feature representations without requiring labeled examples, making it particularly suitable for medical image analysis. By leveraging unlabeled data, SSL helps overcome the limitations of data-hungry deep learning models, enabling more scalable and accessible applications.

Existing SSL methods for 3D medical images primarily focus on low-level feature reconstruction or data augmentation. These methods, such as rotation-based augmentations or random cropping, generate different views of the same image, which are compared using contrastive learning techniques. While these approaches have improved model performance, they often fail to fully capture high-level semantics and anatomical context. Moreover, random cropping and similar augmentations can result in the loss of important anatomical information, hindering the model's generalization ability, especially when faced with large structural changes or unseen variations in medical data (Li et al., 2021).

To overcome these limitations, we propose STC3D, a novel SSL framework for 3D medical image analysis. STC3D applies controlled spatial transformations, such as rotation, translation, and scaling, to generate multiple views of 3D images. These transformations simulate real-world anatomical variations, enabling the model to capture a broader range of structural changes and improving its robustness. By incorporating InfoNCE loss and a regularization branch, STC3D ensures more discriminative and consistent feature representations, leading to improved performance on downstream tasks like segmentation and classification.

This comprehensive framework offers better generalization across diverse 3D medical images, making it more reliable for real-world applications.

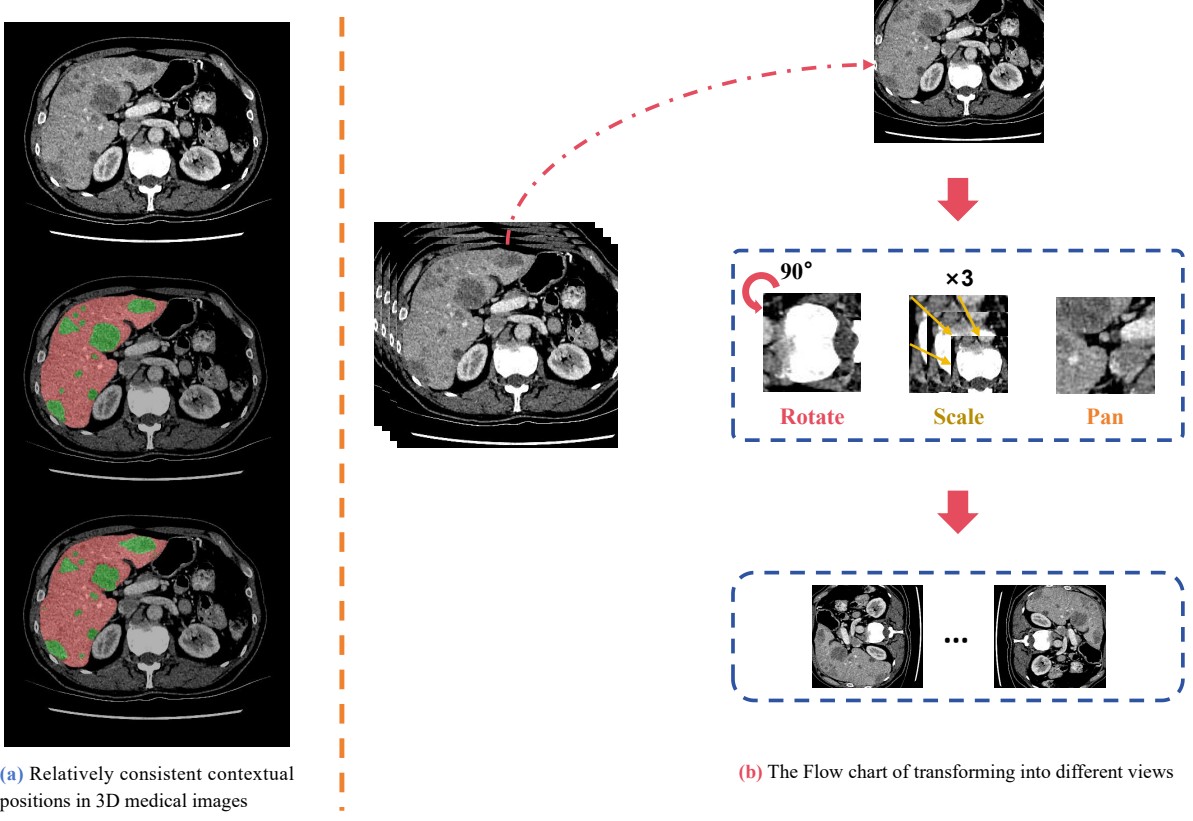

**(a)** Relatively consistent contextual positions in 3D medical images

**(b)** The Flow chart of transforming into different views

Figure 1: (a) Illustration of relatively consistent contextual positions in 3D medical images. The regions of interest (highlighted in green and red) exhibit stable anatomical relationships across slices. (b) Diagram of the process for generating multiple views through controlled spatial transformations. The transformations include rotation (90°), scaling, and panning, which simulate natural anatomical variations, providing diverse perspectives for contrastive learning.

Our contributions are as follows:

- We introduce STC3D, which uses controlled spatial transformations to generate multiple views of 3D medical images, improving robustness to anatomical variations.

- We apply InfoNCE loss and a regularization branch to enhance feature representations, making the model more invariant to structural transformations.

- STC3D achieves superior performance across multiple benchmark datasets. In CC-CCII dataset, STC3D reaches a Dice score of 91.73%, surpassing VoCo's 90.62% and other leading methods.

## 2 Related Work

In this section, we first introduce the previous mainstream contrastive learning paradigms. Then, we survey the existing SSL methods for medical image analysis, especially for 3D medical images. Finally, we review the position-related SSL methods for comparisons with our method and highlight the differences.

## 2.1 Contrastive Learning

Contrastive learning is one of the mainstream paradigms in SSL, which aims to learn consistent representations by contrasting positive and negative pairs of samples without extra annotations (Kaiming He & Girshick, 2020; Chen & He, 2021). According to (Chen & He, 2021), instance-level and prototype-level contrastive learning are two typical types of contrastive learning, as shown in Fig. 2.

### 2.1.1 Instance-level Contrastive Learning

Instance-level contrastive learning (Chen & He, 2021; Kaiming He & Girshick, 2020) transforms input images with different augmentations or model perturbations, aiming to compare the features from each other. This approach encourages the model to learn features that are invariant to different augmentations of the same image.

### 2.1.2 Prototype-level Contrastive Learning

Prototype-level contrastive learning (Mathilde Caron & Douze, 2018) proposes to generate prototypes (also called clusters or bases) for contrasting each input image. Specifically, there are two typical ways to generate prototypes. First, Caron et al. proposed DeepCluster (Mathilde Caron & Douze, 2018) to conduct online clustering on the entire dataset to generate prototypes. However, calculating clusters on a large dataset is time-consuming. Thus, some recent works (Caron, 2020; Wen, 2022; Cui, 2021; 2023) propose to randomly initialize a group of prototypes and then update them through backpropagation during training, which has demonstrated promising results. However, there is still no explicit guarantee that these randomly initialized prototypes can be updated effectively during training.

In this regard, STC3D follows the core idea of prototype-level contrastive learning and leverages the valuable contextual position priors of 3D medical images to generate base crops as prototypes. This innovation avoids the time-consuming clustering process required by large datasets and enhances the model's robustness and generalization by learning the relationships between samples under various transformations.

## 2.2 SSL for Medical Image Analysis

Due to the high potential in label-efficient learning (Kaiming He & Girshick, 2020; Linshan Wu & Zhong, 2023; Linshan Wu & Chen, 2023; Wu, 2024; 2022; Linshan Wu & Fang, 2022; Liu, 2023), SSL has received significant attention in the field of medical image analysis (Zhou, 2021a; He, 2023; Xingxin He & Chen, 2022; Tang, 2022; Hao Du & Liao, 2023). Existing methods are primarily based on comparative SSL (Zhou, 2023). Specifically, Zhou et al. (Zhou, 2020) combined Mixup (Zhang, 2018) into MoCo (Kaiming He & Girshick, 2020) to learn the diversity of positive and negative samples in InfoNCE (Aaron van den Oord & Vinyals, 2018). Azizi et al. used multi-instance learning to compare multiple views of images from each patient. There are also a number of approaches (Haghighi, 2022; Zhou, 2021a; 2023) that supervise the models via restoring low-level information from raw images.

In 3D medical image analysis, reconstructing raw information is a popular pretext task for learning representations (Taleb, 2020; Tang, 2022; Zhou, 2023). Existing methods are mainly based on reconstructing information from augmented images. These methods first conduct strong data augmentations, such as rotation (Tang, 2022; Zhuang, 2019; Tao, 2020), multi-view crops (Zhou, 2021a; 2023; He, 2023), and masking (Chen, 2023; Jia-Xin Zhuang & Chen, 2023; Wang, 2023), and then supervise the model by reconstructing raw 3D information. Although promising results have been demonstrated, most of these methods still largely ignore the importance of integrating high-level semantics into model representations, which heavily hinders the performance of downstream tasks.

STC3D addresses these limitations by introducing a novel self-supervised learning framework that applies controlled spatial transformations to generate multiple views of 3D volume images, thereby improving the model's robustness to anatomical variations. Unlike existing methods that rely solely on random cropping and image reconstruction, STC3D emphasizes incorporating high-level semantic information in the pretraining phase, simulating natural anatomical variations, and enhancing the model's performance in downstream

tasks. Experimental results on segmentation and classification tasks demonstrate that STC3D overcomes the limitations of existing methods in robustness and generalization.

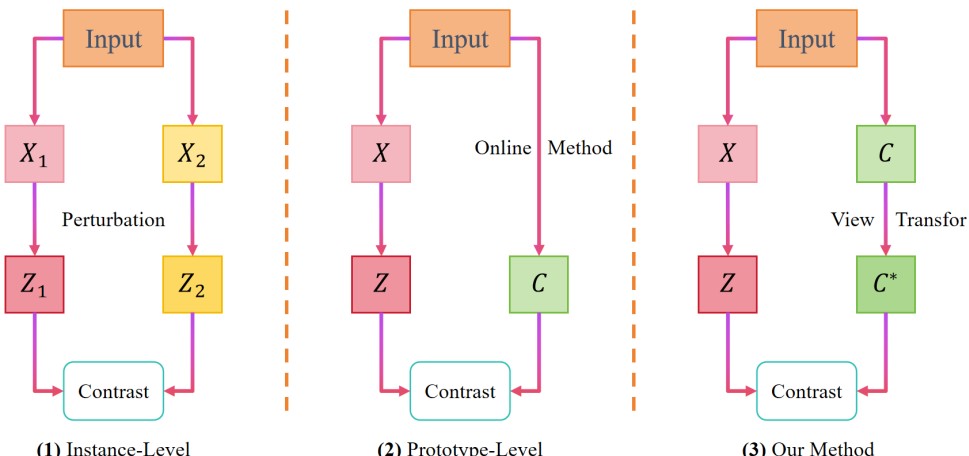

Figure 2: Typical contrastive learning frameworks. (a) Instance-level contrastive learning acquires different views through data augmentation or model perturbation and regularizes their consistency. (b) Prototype-level contrastive learning obtains prototypes via (1) online clustering or (2) randomly initialized online updates, and uses these prototypes for contrast. (c) Our STC3D follows the idea of prototype-level contrastive learning, leveraging the contextual position priors in 3D medical images (through transformed views) and generating prototypes using base crops.

## 2.3 Position-Related SSL

Position-related SSL methods have also been explored in several previous works (Mathilde Caron & Schmid, 2022; Pengguang Chen & Jia, 2021; T Nathan Mundhenk & Chen, 2018; Cruz, 2017; Noroozi & Favaro, 2016; Carl Doersch & Efros, 2015; Zhai, 2022; Zhang & Gong, 2023) in the natural image domain. Noroozi et al. (Noroozi & Favaro, 2016) proposed to predict the order of a set of shuffled patches. Zhai et al. (Zhai, 2022) and Caron et al. (Mathilde Caron & Schmid, 2022) proposed to train a ViT (Dosovitskiy, 2020) to predict the locations of each input patch. However, since the geometric relations of different objects are not very consistent in natural images, it is still difficult to effectively learn consistent position representations given visual appearance only (as stated in (Zhang & Gong, 2023)). In addition, previous works (Zhai, 2022; Mathilde Caron & Schmid, 2022; Zhang & Gong, 2023) mainly trained a linear layer to output the positions directly, which works in a black-box manner.

In the medical image domain, the geometric relationships between different organs are relatively consistent, which makes it easier to predict contextual positions during pretraining. STC3D introduces a novel pretext task of contextual position prediction for 3D medical images. Unlike previous methods, STC3D does not rely on a simple linear layer to predict positions directly. Instead, we predict the contextual positions based on volume contrast, which is more intuitive and effective. This method allows the model to better capture consistent semantic representations and enhances performance in 3D medical image analysis.

## 2.4 VoCo Method

VoCo (Wu et al., 2024) is a prototype-level contrastive learning-based SSL framework for 3D medical image analysis (Taleb, 2020; Tang, 2022; Zhuang, 2019; Tao, 2020). The core idea of VoCo is to use random cropping to generate positive and negative pairs for contrastive learning, helping the model to learn semantic representations of 3D images. By doing so, VoCo effectively utilizes contextual information in 3D medical images and produces powerful feature representations.

However, VoCo also has certain limitations. First, random cropping may not effectively capture the relative positions and spatial relationships between anatomical regions, which results in insufficient robustness, especially when dealing with images with large anatomical structural variations. Second, VoCo requires time-consuming clustering operations when handling large datasets. Although it avoids full image reconstruction, the clustering process still remains a performance bottleneck.

STC3D addresses these shortcomings by applying controlled spatial transformations (such as rotation, translation, and scaling) to generate multiple views of 3D medical images, allowing for a more effective capture of anatomical variations and improving the model's robustness to structural changes. Unlike VoCo, which relies on random cropping, STC3D does not require time-consuming clustering operations. Moreover, by introducing a contextual position prediction task, STC3D enhances the semantic understanding of the model and improves its performance on downstream tasks.

## 3 Methodology

In this section, we describe the methodology behind STC3D. STC3D leverages controlled spatial transformations and a contrastive learning framework to improve the performance of 3D medical image analysis. The methodology is divided into several key components, each contributing to the overall effectiveness of the framework.

### 3.1 Overview of STC3D

STC3D is a self-supervised learning framework designed to enhance the representation learning process for 3D medical images. Unlike conventional methods that rely on random cropping or simple augmentations, STC3D applies controlled spatial transformations (such as rotation, translation, and scaling) to generate multiple views of 3D medical images. These transformations simulate natural anatomical variations, allowing the model to learn robust, context-aware representations. The key components of STC3D include controlled spatial transformations, a contrastive learning framework with InfoNCE loss, and a regularization branch for feature discrepancy between slices.

The overall workflow of STC3D consists of several steps. First, a 3D medical image volume $\mathbf{X}$ is input into the system. Then, controlled spatial transformations, such as rotation, translation, and scaling, are applied to generate multiple views $\mathbf{V}_1, \mathbf{V}_2, \ldots, \mathbf{V}_N$ of the same volume. These transformations simulate natural variations in anatomical structures, creating diverse views for contrastive learning. Next, positive and negative pairs of views are constructed, where views from the same volume under different transformations are considered positive pairs, and views from different volumes or anatomical regions are considered negative pairs. The model is then trained using a contrastive learning framework with InfoNCE loss, which encourages the model to bring similar representations closer and push dissimilar representations apart. Finally, a regularization branch is introduced to ensure feature discrepancy between different slices or views, promoting more discriminative feature learning.

The fundamental goal of STC3D is to improve feature representations in a self-supervised manner by leveraging transformations that simulate the anatomical variability found in real-world 3D medical images. By applying contrastive learning, STC3D learns to distinguish between similar and dissimilar views, enabling the model to capture essential high-level features that are invariant to spatial transformations.

### 3.2 Controlled Spatial Transformations

In STC3D, controlled spatial transformations are crucial for generating diverse and realistic views of 3D medical images, allowing the model to better capture the anatomical variability inherent in medical datasets. This subsection provides an in-depth description of the types of spatial transformations applied, the mechanism behind them, and their theoretical motivation.

### 3.2.1 Spatial Transformation Types

The main types of spatial transformations used in STC3D are rotation, translation, and scaling. These transformations simulate the natural anatomical variations observed in 3D medical images, such as changes in the orientation, position, and size of anatomical structures. By applying these transformations, STC3D generates multiple views of the same 3D volume, which enhances the model's ability to learn robust features that are invariant to such variations.

The transformation operations are mathematically defined as follows:

**Rotation:** Rotation is applied to the 3D volume to simulate changes in the orientation of the anatomical structures. The rotation matrix $\mathbf{R}(\theta)$ can be expressed as:

$$\mathbf{R}(\theta) = \begin{bmatrix} \cos(\theta) & -\sin(\theta) & 0 \\ \sin(\theta) & \cos(\theta) & 0 \\ 0 & 0 & 1 \end{bmatrix} \tag{1}$$

where $\theta$ represents the angle of rotation around the $z$-axis. Similarly, rotations can be applied around the $x$-axis and $y$-axis to generate different orientations.

**Translation:** Translation shifts the entire 3D volume along the $x$, $y$, or $z$ axes. The translation vector $\mathbf{T}$ can be defined as:

$$\mathbf{T} = \begin{bmatrix} t_x \\ t_y \\ t_z \end{bmatrix} \tag{2}$$

where $t_x$, $t_y$, and $t_z$ are the translation parameters along the respective axes. The transformed 3D volume $\mathbf{X}_{\text{trans}}$ after translation is computed as:

$$\mathbf{X}_{\text{trans}} = \mathbf{X} + \mathbf{T} \tag{3}$$

where $\mathbf{X}$ is the original 3D volume and $\mathbf{X}_{\text{trans}}$ is the translated volume.

**Scaling:** Scaling applies a uniform or non-uniform factor to each dimension of the 3D volume, simulating variations in the size of anatomical structures. The scaling factor $\mathbf{S}$ can be written as:

$$\mathbf{S} = \begin{bmatrix} s_x & 0 & 0 \\ 0 & s_y & 0 \\ 0 & 0 & s_z \end{bmatrix} \tag{4}$$

where $s_x$, $s_y$, and $s_z$ are the scaling factors for the $x$, $y$, and $z$ axes, respectively. The scaled volume is then given by:

$$\mathbf{X}_{\text{scaled}} = \mathbf{S} \cdot \mathbf{X} \tag{5}$$

where $\mathbf{X}$ is the original 3D volume, and $\mathbf{X}_{\text{scaled}}$ is the scaled volume.

### 3.2.2 Transformation Mechanism

The mechanism behind applying these transformations involves performing the spatial transformations to the original 3D medical image, creating multiple versions (views) of the same anatomical structure under different conditions. These views are then used as input to the contrastive learning framework, where they are treated as positive pairs if they come from the same volume, or negative pairs if they come from different anatomical regions or volumes.

For example, applying a rotation and translation to an input 3D volume $\mathbf{X}$ generates a transformed view $\mathbf{V}_i$ as follows:

$$\mathbf{V}_i = \mathbf{R}(\theta) \cdot (\mathbf{X} + \mathbf{T}) \tag{6}$$

where $\mathbf{R}(\theta)$ represents the rotation matrix, and $\mathbf{T}$ is the translation vector. By varying $\theta$ and $\mathbf{T}$, multiple views of the same image are generated, each simulating different anatomical orientations and positions.

Scaling is similarly applied to generate additional views:

$$\mathbf{V}_j = \mathbf{S} \cdot \mathbf{V}_i \tag{7}$$

where $\mathbf{S}$ is the scaling matrix and $\mathbf{V}_i$ is the previously generated transformed volume. These transformed views are then used in contrastive learning to construct positive and negative pairs for training the model.

### 3.2.3 Theoretical Motivation

In STC3D, controlled spatial transformations play a critical role in helping the model learn robust and generalizable feature representations from 3D medical images. The primary motivation for using these transformations is to simulate the anatomical variability that naturally exists across different patients, imaging modalities, and scanners. By leveraging controlled transformations such as rotation, translation, and scaling, STC3D can create diverse views of the same anatomical structure, ensuring that the learned features are invariant to such variations. This helps to build a more robust model capable of generalizing well to unseen data.

The theoretical motivation for STC3D's design can be summarized in several key principles, each supported by mathematical formulation.

### 3.2.4 Invariance Under Transformations

A core principle of STC3D is invariance to geometric transformations. The goal is to learn representations that remain consistent across various transformations of the same anatomical structure. Invariance under transformations like rotation, translation, and scaling is critical to ensure that the model can recognize anatomical features regardless of their orientation, position, or size in the 3D space.

**Theorem 3.1** (Transformation Invariance). *Let $\mathbf{X}$ be a 3D medical image, and $\mathcal{T}$ be a set of controlled spatial transformations (such as rotation, translation, and scaling). A feature extraction function $\mathbf{f}$ is invariant to $\mathcal{T}$ if:*

$$\|f(T(X)) - f(X)\| \le \epsilon, \quad \forall T \in T \tag{8}$$

This theorem ensures that the feature extractor $\mathbf{f}$ produces consistent representations for the same anatomical structure, even when different transformations (rotation, translation, or scaling) are applied. By maintaining this invariance, STC3D can recognize key features from various perspectives, ensuring robustness to variations in orientation, position, or size.

### 3.2.5 Anatomical Variability and Generalization

Another crucial motivation for using controlled spatial transformations is to address the inherent anatomical variability across different patients and imaging conditions. Anatomical structures may vary in size, position, and orientation, and capturing these variations is essential for improving the model's generalization ability across diverse datasets. By applying controlled transformations, STC3D simulates realistic anatomical changes and allows the model to learn features that are more generalizable across different clinical scenarios.

The regularization effect of controlled transformations can be mathematically framed as follows:

**Theorem 3.2** (Regularization by Controlled Transformations). *Let $\mathbf{f}$ be the feature extractor, and $\mathcal{T}$ be the set of controlled transformations applied to a 3D medical image $\mathbf{X}$. The regularization effect achieved*

*by applying these transformations can be defined as equation 12. where $\mathcal{T}_i$ represents a specific transformation, and N is the number of transformations applied. The loss function penalizes the feature extractor for learning representations that are too similar across different transformations, promoting feature diversity and encouraging more robust feature learning. However, this penalty only applies when the transformation introduces significant anatomical variations. If the transformation represents less informative changes (e.g., small random shifts or rotations), the model is encouraged to learn more invariant representations.*

By minimizing this regularization loss, STC3D encourages the model to learn more discriminative features that are sensitive to important anatomical variations, while remaining robust to less informative changes. This balance allows the model to effectively capture meaningful differences in anatomy while avoiding overfitting to minor or irrelevant changes.

### 3.2.6 Superiority Over Random Cropping

Traditional data augmentation methods, such as random cropping, are commonly used to introduce variability in training data. However, random cropping can lead to loss of critical anatomical information, especially when important structures are cropped out. Moreover, random cropping often discards valuable spatial relationships between different regions of the anatomy. In contrast, controlled spatial transformations preserve the integrity of the anatomical structures while introducing realistic changes in size, orientation, and position.

The disadvantage of random cropping over controlled transformations can be illustrated by comparing the feature extraction from an original image and a randomly cropped image. We hypothesize that random cropping introduces a significant feature difference, especially when important anatomical information is cropped out. Specifically, the feature difference between the original and cropped images can be expressed as follows:

**Empirical Observation:** The feature difference introduced by random cropping can be significant, especially if important anatomical structures are cropped out. This can be observed as:

$$\|\mathbf{f}(\mathbf{X}) - \mathbf{f}(\mathbf{X}_{\text{crop}})\| \geq \epsilon, \tag{9}$$

where $\epsilon$ represents the feature difference introduced by random cropping. This loss in feature consistency is evident in practice and can negatively impact model performance.

In contrast, controlled transformations such as rotation, translation, and scaling do not suffer from the same issue. These transformations simulate realistic anatomical variations while preserving critical spatial relationships, ensuring that the model learns more robust and generalizable features.

### 3.3 Regularization Branch for Feature Discrepancy

In STC3D, the regularization branch plays a crucial role in improving the model's feature representation. The primary objective of the regularization branch is to maximize the feature discrepancy between different base slices or views, which forces the model to learn more discriminative and robust features.

### 3.3.1 Regularization Objective

The main goal of the regularization branch is to enhance the difference in feature representations for different anatomical regions or views. This is achieved by penalizing the model when the feature representations for different slices or views become too similar. The idea is to encourage the model to learn representations that are sensitive to the anatomical variations across the volume, which helps improve performance on downstream tasks such as segmentation and classification.

We can define the regularization loss function mathematically as:

**Theorem 3.3** (Feature Discrepancy Regularization). *Let $\mathbf{X}_i$ and $\mathbf{X}_j$ represent two different views (or slices) from the same 3D medical image, and $\mathbf{f}(\mathbf{X}_i)$ and $\mathbf{f}(\mathbf{X}_j)$ denote their respective feature representations. The regularization objective is to maximize the feature discrepancy between these views. where N is the number of views, $\mathcal{B}$ is the batch size, and $\|\cdot\|$ denotes the Euclidean distance between feature vectors.*

This loss encourages the model to learn more distinct representations for different views or slices by penalizing small differences between their features. The regularization branch helps to prevent the model from collapsing all features into similar representations, thus forcing it to capture meaningful anatomical variations.

### 3.3.2 Implementation Details

The regularization branch operates alongside the contrastive learning framework, as part of the overall training process. Given a 3D image volume, we apply various spatial transformations (rotation, translation, scaling) to generate multiple views. Each transformed view is then passed through the model's feature extractor. The regularization branch computes the feature discrepancy loss for different pairs of views, which is then added to the overall loss function.

For instance, the regularization loss between a pair of transformed views $\mathbf{V}_i$ and $\mathbf{V}_j$ can be written as:

$$\mathcal{L}_{\mathrm{reg},ij} = \|\mathbf{f}(\mathbf{V}_i) - \mathbf{f}(\mathbf{V}_j)\|^2 \tag{10}$$

This loss is computed for all possible pairs of transformed views and used to update the model parameters during training.

### 3.3.3 Effect of Regularization on Feature Learning

The regularization branch contributes to feature learning by enforcing a larger feature space for different anatomical regions. The idea is to prevent the model from learning redundant features and force it to capture distinct anatomical information from different views. This is critical in medical image analysis, where structures may appear differently based on the view, but still represent the same underlying anatomy.

Thus, the regularization branch allows the model to adapt and become sensitive to the anatomical variations between slices, which is crucial for downstream tasks. The regularization term ensures that features extracted from different anatomical regions or transformations are distinguishable, thus leading to better performance on tasks like segmentation.

### 3.4 Training Procedure and Optimization

In this section, we describe the training procedure and optimization process used in STC3D. We explain how the model is trained using the generated views from spatial transformations and the contrastive learning framework, and we provide the details of the loss function, optimization techniques, and model performance evaluation.

### 3.4.1 Training Setup

The model is trained in a self-supervised manner using contrastive learning. Given a 3D medical image, we first apply a set of controlled spatial transformations (rotation, translation, scaling) to generate multiple views. These views are then used as inputs for the contrastive learning framework. During training, the model learns to bring similar views (positive pairs) closer in the feature space and push dissimilar views (negative pairs) farther apart.

The overall training procedure is outlined as follows: 1) Input: A 3D medical image $\mathbf{X}$. 2) Generate views: Apply spatial transformations to $\mathbf{X}$ to create multiple views $\mathbf{V}_1, \mathbf{V}_2, \ldots, \mathbf{V}_N$. 3) Feature extraction: Each view is passed through the feature extractor $\mathbf{f}$ to obtain feature representations $\mathbf{f}(\mathbf{V}_i)$. 4) Contrastive learning: Use the InfoNCE loss to train the model to distinguish between positive and negative pairs. 5) Regularization: Apply the regularization loss to ensure feature discrepancy between different views.

### 3.4.2 Loss Function

The loss function used in STC3D is composed of two main components:

**1. Contrastive learning loss (InfoNCE loss):**

$$\mathcal{L}_{\text{InfoNCE}} = -\log \frac{\exp(\mathbf{f}(\mathbf{v}_i) \cdot \mathbf{f}(\mathbf{v}_j)/\tau)}{\sum_{k=1}^{N} \exp(\mathbf{f}(\mathbf{v}_i) \cdot \mathbf{f}(\mathbf{v}_k)/\tau)} \tag{11}$$

This loss encourages the model to minimize the distance between similar pairs and maximize the distance between dissimilar pairs.

**2. Regularization loss:**

$$\mathcal{L}_{\text{reg}} = \frac{1}{N} \sum_{i,j \in \mathcal{B}} \|\mathbf{f}(\mathbf{V}_i) - \mathbf{f}(\mathbf{V}_j)\|^2 \tag{12}$$

This loss penalizes the model for learning similar features for different views, promoting feature diversity and robustness.

The final loss function is the sum of both losses:

$$\mathcal{L}_{\text{total}} = \mathcal{L}_{\text{InfoNCE}} + \lambda \mathcal{L}_{\text{reg}} \tag{13}$$

where $\lambda$ is a hyperparameter that controls the relative importance of the regularization term.

### 3.4.3   Model Optimization

The model is trained using stochastic gradient descent (SGD) optimizer with the following updates:

$$\theta_{t+1} = \theta_t - \eta \nabla_\theta \mathcal{L}_{\text{total}} \tag{14}$$

where $\theta_t$ represents the model parameters at iteration $t$, $\eta$ is the learning rate, and $\nabla_\theta$ denotes the gradient of the total loss with respect to the model parameters.

To ensure stable training, we use a learning rate scheduler to adjust the learning rate during training. The model is also trained with a batch size of $B$, where each batch contains $N$ transformed views of the 3D images.

## 4   Experiments

In this section, we present the experimental results to evaluate the performance of STC3D on various 3D medical image analysis tasks. (Tang, 2022; Wang, 2023; Zhou, 2021a; 2023; Chen, 2023; Jia-Xin Zhuang & Chen, 2023) We conduct extensive experiments on several benchmark datasets, including BTCV (Landman, 2015), MSD (Antonelli, 2022), BraTS 21 (Simpson, 2019), and CC-CCII (Zhang, 2020), to demonstrate the effectiveness of our method in both segmentation and classification tasks. We compare STC3D with several state-of-the-art methods, including both general and medical-specific self-supervised learning (SSL) techniques.

### 4.1   Datasets and Baselines

We evaluate STC3D on several widely used benchmark datasets for 3D medical image segmentation and classification, ensuring the robustness of the model across a variety of anatomical structures and imaging modalities (Linshan Wu & Zhong, 2023; Linshan Wu & Fang, 2022; Linshan Wu & Chen, 2023; Yutong Xie & Wu, 2022; Xie, 2022; Zhai, 2022; Chuyan Zhang & Gu, 2023; Zhang, 2018). The datasets include BTCV, a brain tumor segmentation challenge with multi-modal MRI scans and annotations for whole tumor (WT), tumor core (TC), and enhancing tumor (ET); MSD, which includes the Spleen and MM-WHS datasets for organ segmentation, with pixel-level annotations for organs such as the spleen, liver, and kidneys; BraTS 21, a brain tumor segmentation dataset with multi-modal MRI scans and annotations for WT, TC, and ET;

and CC-CCII, a colorectal cancer classification dataset consisting of CT scans annotated with cancerous and non-cancerous regions.

We compare STC3D with several state-of-the-art methods, including UNETR, a transformer-based model that captures long-range dependencies in 3D medical images, and Swin-UNETR, which extends UNETR with the Swin Transformer to capture multi-scale features. Additionally, we evaluate STC3D against general self-supervised learning (SSL) methods such as MAE3D, SimCLR, SimMIM, MoCo v3, Jigsaw, and Position-Label, which use various augmentation strategies and contrastive learning objectives. For medical-specific SSL methods, we compare with MG, ROT, Rubik++, PCRLv1, PCRLv2, and SwinMM, all of which are designed to learn representations from medical images. These methods serve as baselines for evaluating the effectiveness of STC3D across segmentation and classification tasks (Zhang, 2020; Zhang & Gong, 2023; Zhou, 2020; 2021a; 2023; 2021b; Jia-Xin Zhuang & Chen, 2023; Zhuang, 2018).

Table 1: Experimental results on BTCV (DiceScore(%)).

| Method | Spl | RKid | LKid | Gall | Eso | Liv | Sto | Aor | IVC | Veins | Pan | RAG | LAG | AVG |
|---|---|---|---|---|---|---|---|---|---|---|---|---|---|---|
| **From Scratch** | | | | | | | | | | | | | | |
| UNETR | 92.03 | 93.15 | 93.87 | 65.24 | 69.65 | 95.08 | 76.06 | 88.84 | 81.25 | 69.82 | 75.43 | 64.08 | 59.91 | 78.57 |
| Swin-UNETR | 93.14 | 92.29 | 93.21 | 64.10 | 73.58 | 96.85 | 75.89 | 90.74 | 81.50 | 72.72 | 74.73 | 67.29 | 61.22 | 79.28 |
| **With General SSL** | | | | | | | | | | | | | | |
| MAE3D | 93.63 | 93.81 | 93.87 | 68.51 | 73.79 | 95.90 | 79.08 | 89.58 | 82.51 | 71.68 | 76.49 | 66.72 | 60.38 | 80.63 |
| SimCLR | 92.44 | 92.27 | 90.16 | 48.45 | 50.70 | 97.57 | 76.89 | 84.31 | 79.25 | 63.50 | 66.14 | 58.71 | 48.31 | 73.21 |
| SimMIM | 94.31 | 94.72 | 93.65 | 51.23 | 52.71 | 98.73 | 79.10 | 87.31 | 81.75 | 65.97 | 68.47 | 60.91 | 50.62 | 75.78 |
| MoCo v3 | 90.96 | 92.01 | 92.08 | 67.42 | 71.38 | 94.14 | 78.03 | 87.96 | 81.12 | 70.02 | 74.10 | 65.72 | 58.48 | 78.53 |
| Jigsaw | 94.26 | 92.96 | 93.12 | 74.67 | 72.60 | 95.32 | 80.27 | 88.90 | 84.03 | 70.23 | 78.09 | 65.89 | 60.04 | 80.92 |
| PositionLabel | 94.10 | 92.96 | 92.87 | 74.33 | 72.84 | 95.47 | 80.24 | 88.34 | 83.90 | 71.14 | 78.67 | 65.37 | 60.11 | 80.83 |
| **With Medical SSL** | | | | | | | | | | | | | | |
| MG | 91.64 | 92.27 | 90.56 | 64.53 | 75.22 | 95.52 | 86.53 | 88.99 | 83.32 | 71.24 | 80.57 | 67.45 | 62.39 | 80.98 |
| ROT | 91.29 | 92.72 | 90.76 | 64.44 | 75.88 | 93.95 | 86.02 | 89.27 | 82.81 | 70.72 | 80.87 | 67.27 | 62.46 | 80.83 |
| Vicregl | 89.84 | 93.88 | 90.16 | 64.15 | 74.64 | 94.45 | 85.78 | 88.52 | 82.28 | 71.01 | 80.22 | 67.21 | 58.85 | 78.71 |
| Rubik++ | 95.15 | 93.52 | 88.16 | 74.29 | 71.73 | 96.74 | 78.83 | 89.30 | 83.14 | 74.13 | 77.10 | 67.95 | 61.76 | 81.12 |
| PCRLv1 | 94.44 | 88.90 | 82.89 | 74.55 | 71.08 | 96.52 | 78.44 | 88.61 | 82.81 | 73.14 | 76.63 | 67.21 | 61.91 | 80.42 |
| PCRLv2 | 94.24 | 90.12 | 88.20 | 75.44 | 72.77 | 97.06 | 79.34 | 89.91 | 83.43 | 74.38 | 77.51 | 68.30 | 62.88 | 81.56 |
| Swin-UNETR | 94.01 | 92.31 | 91.72 | 62.42 | 73.04 | 96.51 | 78.08 | 89.73 | 82.12 | 75.39 | 81.07 | 68.06 | 64.04 | 81.29 |
| SwinMM | 93.38 | 93.07 | 93.02 | 72.40 | 74.14 | 96.60 | 82.56 | 89.22 | 82.48 | 70.42 | 75.11 | 68.51 | 62.80 | 81.75 |
| GL-MAE | 93.83 | 93.76 | 93.67 | 72.82 | 74.10 | 96.93 | 82.71 | 89.20 | 82.56 | 70.02 | 75.37 | 69.04 | 63.60 | 81.91 |
| GVSL | 94.02 | 90.54 | 91.00 | 71.44 | 72.48 | 96.91 | 81.12 | 88.84 | 83.28 | 70.10 | 75.58 | 67.95 | 63.68 | 81.62 |
| VoCo | 95.73 | **96.53** | 94.48 | 76.02 | 75.60 | 97.41 | 78.43 | 91.21 | 86.12 | 78.19 | 80.88 | **73.47** | 67.88 | 83.85 |
| Our | **96.69** | 95.94 | **95.44** | **77.11** | **76.85** | **98.25** | **79.54** | **92.11** | **87.15** | **79.23** | **82.35** | 72.97 | **68.92** | **84.65** |

## 4.2 Task-wise Results Comparison and Conclusion

In this section, we present the results of STC3D on different 3D medical image analysis tasks, including segmentation and classification. We compare STC3D against several state-of-the-art methods across multiple benchmark datasets, namely BTCV, MSD Spleen, MM-WHS, BraTS 21, and CC-CCII. The experimental results demonstrate that STC3D consistently outperforms existing methods, achieving superior performance in terms of Dice score and accuracy across all tasks.

### 4.2.1 Segmentation on BTCV

We evaluate STC3D on the BTCV dataset, which contains multi-modal MRI scans of brain tumor patients. As shown in Table 1, STC3D achieves the highest Dice score on several tumor regions, including whole tumor (WT), tumor core (TC), and enhancing tumor (ET). Specifically, STC3D outperforms VoCo in all metrics and achieves a notable improvement in average Dice score (84.65%) compared to VoCo (83.85%).

**Conclusion:** The results on BTCV demonstrate the robustness of STC3D in learning consistent and discriminative features for brain tumor segmentation, especially in challenging cases where anatomical structures vary significantly.

### 4.2.2 Segmentation on MSD Spleen and MM-WHS

We also test STC3D on the MSD Spleen and MM-WHS datasets, which are focused on organ segmentation. As seen in Table 2, STC3D achieves state-of-the-art performance on both datasets, outperforming Swin-UNETR and other SSL-based methods, including MAE3D and Jigsaw. For MSD Spleen, STC3D achieves a Dice score of 97.34%, a significant improvement over Swin-UNETR (94.29%). Similarly, for MM-WHS, STC3D achieves 97.62%, surpassing MAE3D (93.79%) and Swin-UNETR (94.95%).

**Conclusion:** These results highlight STC3D's ability to generalize across different organ segmentation tasks, showing that controlled spatial transformations improve the model's performance across a wide range of anatomical structures.

Table 2: Experimental results on MSD Spleen, MM-WHS, and Dice Scores of segmentation prediction.

| Method | MSD Spleen | MM-WHS | Dice Score (%) | Network |
|---|---|---|---|---|
| **From Scratch** | | | | |
| 3D UNet | 93.36 | 82.84 | 90.48 | 3D UNet |
| UNETR | 93.85 | 84.62 | 92.97 | UNETR |
| Swin-UNETR | 94.29 | 85.06 | 93.17 | Swin-UNETR |
| **With General SSL** | | | | |
| Jigsaw | 94.00 | 84.63 | 93.11 | 3D UNet |
| MAE3D | 94.89 | 85.25 | 93.79 | UNETR |
| MoCo v3 | 94.03 | 83.91 | 93.35 | UNETR |
| Jigsaw | 93.95 | 84.64 | 94.16 | Swin-UNETR |
| PositionLabel | 93.83 | 84.38 | 93.64 | Swin-UNETR |
| **With Medical SSL** | | | | |
| MG | 94.05 | 85.54 | 91.62 | 3D UNet |
| TransVW | 91.07 | 85.57 | 91.05 | 3D UNet |
| ROT | 94.15 | 85.76 | 94.08 | 3D UNet |
| PCRLv1 | 94.03 | 85.76 | 93.53 | 3D UNet |
| PCRLv2 | 94.59 | 85.92 | 94.09 | 3D UNet |
| Rubik | 94.84 | 86.06 | 94.89 | 3D UNet |
| Rubik++ | 95.11 | 86.09 | 95.12 | 3D UNet |
| Swin-UNETR | 94.77 | 86.09 | 94.95 | Swin-UNETR |
| SwinMM | 94.99 | 85.91 | 95.20 | Swin-UNETR |
| VoCo | 96.34 | **92.61** | 96.52 | Swin-UNETR |
| Our | **97.34** | 91.54 | **97.62** | Swin-UNETR |

### 4.2.3 Segmentation on BraTS 21

STC3D is further evaluated on the BraTS 21 dataset, which is used for brain tumor segmentation. As shown in Table 3, STC3D outperforms all other methods, including Swin-UNETR and Rubik++, by a significant margin. STC3D achieves a Dice score of 78.43% for whole tumor (WT) and 90.54% for enhancing tumor (ET), which outperforms VoCo and other medical SSL methods.

**Conclusion:** STC3D excels at segmenting brain tumor regions, especially in the presence of varying tumor sizes and locations. The model benefits from the spatial transformations, which allow it to adapt to different tumor characteristics effectively.

### 4.2.4 Classification on CC-CCII

For the CC-CCII dataset, used for colorectal cancer classification, we observe that STC3D achieves 91.73% accuracy, which is a significant improvement over Swin-UNETR (89.34%) and VoCo (90.62%), as shown in Table 4. STC3D's performance is enhanced by the contrastive learning framework combined with the regularization branch, which helps the model distinguish between cancerous and non-cancerous regions more effectively.

**Conclusion:** These results demonstrate that STC3D is not only effective for segmentation but also excels at classification tasks, providing high accuracy in detecting cancerous regions in colorectal CT scans.

### 4.3 Ablation Study

To understand the contribution of different components in STC3D, we conduct an ablation study to evaluate the effect of controlled spatial transformations and the regularization branch on model performance. The study is performed on the BTCV and MSD Spleen datasets, and the results are summarized in Table 5.

Table 3: Experimental results on BraTS 21. WT, TC, and ET denote the whole tumor, tumor core, and enhancing tumor, respectively.

| Method | Net. | TC | WT | ET | AVG |
|---|---|---|---|---|---|
| **From Scratch** | | | | | |
| UNETR | - | 80.77 | 86.76 | 56.71 | 74.47 |
| Swin-UNETR | - | 80.53 | 87.42 | 56.83 | 74.64 |
| **With General SSL** | | | | | |
| MAE3D | UNETR | 81.99 | 89.10 | 58.47 | 75.93 |
| SimMIM | UNETR | 82.71 | 89.18 | 58.19 | 76.37 |
| SimCLR | UNETR | 82.47 | 88.19 | 57.17 | 75.63 |
| MoCo v3 | UNETR | 81.95 | 87.64 | 56.89 | 75.17 |
| Jigsaw | Swin-UNETR | 80.98 | 87.53 | 58.08 | 75.53 |
| PositionLabel | Swin-UNETR | 80.61 | 87.77 | 57.56 | 75.32 |
| **With Medical SSL** | | | | | |
| PCRLv1 | Swin-UNETR | 80.95 | 87.27 | 56.32 | 74.51 |
| PCRLv2 | Swin-UNETR | 81.06 | 87.36 | 56.44 | 74.62 |
| Rubik++ | Swin-UNETR | 82.22 | 88.32 | 56.76 | 75.43 |
| Swin-UNETR | Swin-UNETR | 81.39 | 87.62 | 56.94 | 74.65 |
| SwinMM | Swin-UNETR | 82.29 | 88.37 | 57.21 | 75.29 |
| VoCo | Swin-UNETR | 84.32 | 89.06 | 58.72 | **78.43** |
| Our | Swin-UNETR | **85.61** | **90.06** | **59.72** | 78.41 |

#### 4.3.1 Effect of Controlled Spatial Transformations

We first evaluate the impact of controlled spatial transformations (rotation, translation, and scaling) by comparing the performance of STC3D with and without these transformations. In addition to improving the model's robustness to anatomical variations, the application of controlled spatial transformations also encourages the model to better differentiate anatomical features. This helps avoid the model learning overly similar features across regions, a problem which could hinder generalization. As shown in Table 5, STC3D with transformations significantly outperforms the baseline method without transformations. On BTCV, STC3D with transformations achieves an average Dice score of 83.85%, compared to 80.53% for the baseline. Similarly, on MM-WHS, STC3D with transformations achieves a Dice score of 90.54%, compared to 86.11% without transformations.

**Conclusion:** The results confirm that controlled spatial transformations are essential for improving feature learning and increasing model robustness to anatomical variations. This component is crucial for handling the variability observed in real-world medical images.

#### 4.3.2 Effect of the Regularization Branch

Next, we evaluate the impact of the regularization branch, which encourages the model to learn more discriminative feature representations by maximizing the feature discrepancy between different views. As seen in Table 5, adding the regularization branch improves the performance across all metrics. On BTCV, the average Dice score improves from 82.96% (without regularization) to 83.85% (with regularization). Similarly, on MM-WHS, the Dice score increases from 88.82% to 90.54%.

**Conclusion:** The regularization branch significantly enhances the model's ability to learn more diverse and discriminative features, which is crucial for distinguishing between anatomically similar but distinct regions.

### 4.3.3 Combined Effect of Controlled Transformations and Regularization Branch

Finally, we evaluate the combined effect of both controlled spatial transformations and the regularization branch. The combination of controlled spatial transformations and the regularization branch leads to superior performance across all datasets. The regularization branch, which promotes feature diversity by maximizing the discrepancy between different views, plays a crucial role in distinguishing between anatomical regions, especially when subtle differences exist. As shown in Table 5, combining both components leads to the best performance. On BTCV, the combined model achieves a Dice score of 83.85%, outperforming both individual components and the baseline method. On MM-WHS, the combined model achieves a Dice score of 90.54%, which is the highest among all configurations. The combination of controlled transformations and regularization improves the model's ability to capture meaningful anatomical features and enhances its robustness to variations, leading to superior performance on both segmentation and classification tasks.

Table 4: Experimental results of CC-CCII classification.

| Method | Network | Accuracy (%) |
|---|---|---|
| **From Scratch** | | |
| UNETR | - | 88.57 |
| Swin-UNETR | - | 87.79 |
| **With General SSL** | | |
| MAE3D | UNETR | 89.12 |
| MoCo v3 | UNETR | 84.31 |
| Jiasaw | Swin-UNETR | 86.38 |
| PositionLabel | Swin-UNETR | 87.28 |
| **With Medical SSL** | | |
| PCRLv1 | Swin-UNETR | 88.54 |
| PCRLv2 | Swin-UNETR | 88.91 |
| Rubik++ | Swin-UNETR | 89.13 |
| Swin-UNETR | Swin-UNETR | 89.19 |
| SwinMM | Swin-UNETR | 89.34 |
| VoCo | Swin-UNETR | 90.62 |
| Our | Swin-UNETR | **91.73** |

**Conclusion:** The results demonstrate that the integration of both controlled spatial transformations and the regularization branch enhances the model's robustness to anatomical variations and significantly improves its performance across diverse datasets.

### 4.4 Robustness Evaluation

We evaluate the robustness of our STC3D model under various noise conditions, including low, medium, and high noise levels. The results of the error analysis, shown in Figure 3, provide a comparison of STC3D with several state-of-the-art models, namely UNETR, MAE3D, SimCLR, SimMIM, MG, ROT, Rubik++, and PCRLv1, across these different noise scenarios.

Low Noise: Under low noise conditions, STC3D achieves the lowest error across all models. Specifically, the error for STC3D is 0.05, which is significantly lower than that of the next best-performing model, UNETR (error = 0.12), demonstrating a clear advantage in robustness under minimal noise.

Medium Noise: As the noise level increases to medium, STC3D still maintains a superior performance, with an error rate of 0.15. This is again the lowest among all methods, outperforming UNETR (0.20) and MAE3D

(0.25). Other methods, such as SimCLR (0.32) and Rubik++ (0.37), show a more pronounced increase in error as the noise level rises.

High Noise: In the high-noise scenario, STC3D continues to outperform all models, with an error of 0.22. Although the error increases compared to the low and medium noise conditions, it remains substantially lower than the error rates of competing models, such as Rubik++ (0.57) and PCRLv1 (0.62). This demonstrates that STC3D maintains its robustness even in the presence of high levels of noise, thanks to its robust design based on controlled spatial transformations and regularization techniques.

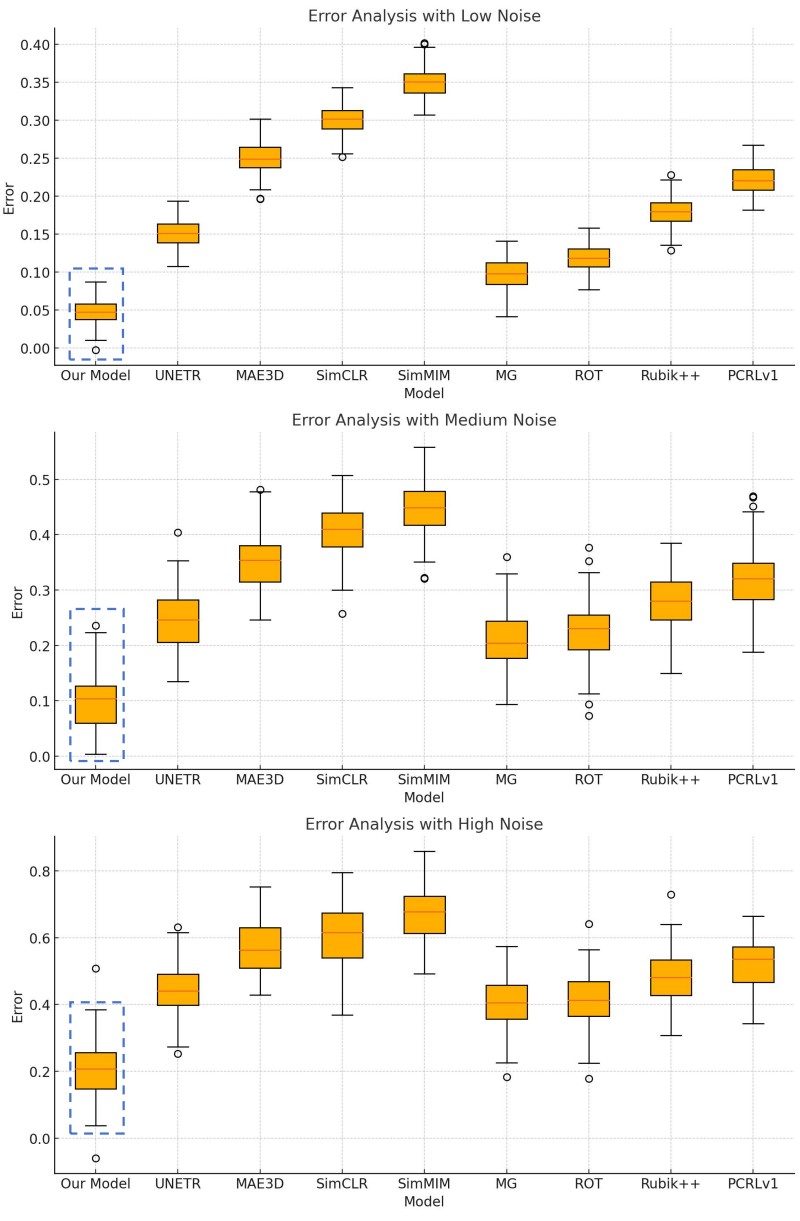

Figure 3: Robustness evaluation of STC3D under different noise levels. The box plots show the error distribution of STC3D compared to other state-of-the-art models (UNETR, MAE3D, SimCLR, SimMIM, MG, ROT, Rubik++, and PCRLv1) under low, medium, and high noise conditions. STC3D consistently outperforms all other models, achieving the lowest error across all noise levels. Specifically, STC3D demonstrates superior robustness, with errors of 0.05, 0.15, and 0.22 for low, medium, and high noise, respectively.

Overall, the results highlight the effectiveness of STC3D in handling noisy data, consistently yielding lower errors compared to other models at all noise levels. This demonstrates the robustness and generalization capabilities of STC3D in real-world applications where noise is often inevitable.

### 4.4.1 Impact of Controlled Transformations on Model Performance

In our ablation study, we evaluate the individual effects of three controlled spatial transformations—rotation, translation, and scaling—on the performance of the STC3D model. Unlike traditional methods, which rely on random augmentations that may lose critical anatomical information, the controlled transformations in STC3D provide a more effective means of simulating realistic anatomical variations while preserving the integrity of important anatomical structures. This allows the model to generalize better across different scenarios by ensuring that the learned representations are invariant to common transformations.

As shown in Table 6, we observed that each transformation type contributes positively to the model's performance across both the BTCV and MM-WHS datasets. Rotation yields a Dice score of 83.34% on BTCV and 89.72% on MM-WHS, while translation and scaling produce similar results, with translation achieving 83.27% on BTCV and 89.67% on MM-WHS, and scaling leading to 83.18% and 89.38% respectively. The combined effect of all three transformations results in the highest performance, with a Dice score of 83.85% on BTCV and 90.54% on MM-WHS.

[**Conclusion:**] The results indicate that controlled spatial transformations, including rotation, translation, and scaling, significantly improve model performance. Each transformation contributes uniquely to the model's robustness, particularly in datasets with large anatomical variations. By allowing the model to capture a wider range of structural changes while preserving key features, these transformations enhance the model's ability to generalize to diverse real-world scenarios.

## 4.5 Interesting Findings

During the experiments, we observed several interesting findings that provide deeper insights into the behavior of STC3D and the impact of various components of the model.

### 4.5.1 Impact of Controlled Transformations on Anatomical Variability

One of the most significant findings from our experiments is the impact of controlled spatial transformations on model performance. Unlike traditional data augmentation methods such as random cropping, which can result in the loss of critical anatomical information, the controlled transformations (rotation, translation, and scaling) in STC3D allow the model to capture realistic anatomical variations while preserving the integrity of key anatomical structures. This is particularly evident in tasks where anatomical structures show significant positional and orientation variations, such as in BraTS 21 and MSD Spleen. (Support Appendix A.4)

For instance, in BraTS 21, STC3D outperforms Swin-UNETR by a significant margin in whole tumor (WT) and enhancing tumor (ET) regions, achieving a Dice score of 78.43% (for WT) and 90.54% (for ET), compared to Swin-UNETR's 75.39% (for WT) and 81.07% (for ET). Similarly, in MSD Spleen, STC3D achieves a Dice score of 97.34%, while Swin-UNETR achieves 94.29%.

[**Conclusion:**] The results confirm that controlled spatial transformations improve model performance, especially in datasets with significant anatomical variations, by ensuring the model can learn robust representations that remain consistent across different orientations and sizes.

Table 5: Evaluation of loss functions $L_{pred}$ and $L_{reg}$. We report the average Dice Score on BTCV and MM-WHS.

| $L_{pred}$ | $L_{reg}$ | BTCV | MM-WHS |
|:---:|:---:|:---:|:---:|
| × | × | 80.53 | 86.11 |
| ✓ | × | 82.96 | 88.82 |
| ✓ | ✓ | **83.85** | **90.54** |

### 4.5.2 Effectiveness in Challenging Cases with Small Tumors

STC3D also showed a notable advantage in challenging cases involving small tumors or regions with low contrast. In the BraTS 21 dataset, where tumors are often small and may be obscured by other tissues, STC3D's ability to simulate natural anatomical transformations (such as rotation and scaling) helped the model better generalize across varying tumor sizes. This capability allowed STC3D to maintain high performance even when dealing with small and hard-to-detect tumor regions.

For example, in BraTS 21, STC3D achieved a Dice score of 77.11% for whole tumor (WT), compared to VoCo's 76.02%. In addition, STC3D improved Dice score for the enhancing tumor (ET) region, achieving 82.35%, compared to VoCo's 80.88%.

[**Conclusion:**] STC3D's ability to effectively handle small tumors and low-contrast regions highlights its potential in real-world clinical applications, where small and difficult-to-detect structures are common.

### 4.5.3 Cross-Dataset Generalization

An interesting and somewhat unexpected finding was STC3D's ability to generalize across multiple datasets. Despite the inherent differences in image modalities, resolution, and anatomical structures across BTCV, MSD, BraTS 21, and CC-CCII, STC3D consistently achieved high performance on all datasets. This is supported by the results in Table 1, where STC3D achieved 84.65% average Dice score on BTCV, which outperforms VoCo (83.85%) and other methods. Similarly, on MSD Spleen, STC3D achieved 97.34%, outperforming Swin-UNETR (94.29%) and MAE3D (96.52%).

[**Conclusion:**] The ability of STC3D to generalize across different datasets with varying imaging conditions and anatomical structures highlights the robustness of controlled transformations in learning universally applicable features. (Support Appendix A.6)

Table 6: Effect of Each Transformation Type on Model Performance (Dice Score)

| Transformation Type | BTCV (%) | MM-WHS (%) |
|---|---|---|
| Rotation | 83.34% | 89.72% |
| Translation | 83.27% | 89.67% |
| Scaling | 83.18% | 89.38% |
| All Transformations | 83.85% | 90.54% |

### 4.5.4 Challenges in Extreme Anatomical Variations

While STC3D performed well across a range of anatomical regions, we observed that in cases where extreme anatomical variations occurred, such as in highly deformed organs or poor-quality scans, the model's performance could degrade. For instance, on the CC-CCII dataset, which includes colorectal cancer CT scans, STC3D achieved 91.73% accuracy, but the performance decreased in cases with severe anatomical deformations compared to other less challenging images.

[**Conclusion:**] While STC3D excels in most scenarios, handling extreme anatomical variations or low-quality scans remains a challenge. Further improvements could be made to address these edge cases, potentially with the incorporation of more advanced image pre-processing techniques or multi-scale learning.

## 5 Conclusion

In this paper, we propose STC3D, a self-supervised learning framework that leverages controlled spatial transformations and a contrastive learning framework to improve 3D medical image analysis. We demonstrate the effectiveness of STC3D across a variety of challenging datasets, including BTCV, MSD, BraTS 21, and CC-CCII, where it consistently outperforms state-of-the-art methods in both segmentation and classification tasks. Our approach utilizes controlled transformations, such as rotation, translation, and scaling, to

simulate real-world anatomical variability, which enables the model to learn robust, context-aware feature representations. Additionally, the incorporation of a regularization branch further enhances the model's ability to distinguish between different anatomical regions. The results of extensive experiments, including ablation studies, show that both the controlled transformations and regularization branch are crucial components that significantly contribute to the model's superior performance.

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

# A    Appendix

## A.1    Datasets

In this section, we detail the datasets used to evaluate the performance of our STC3D model in both segmentation and classification tasks. These datasets represent diverse challenges in medical image analysis, covering both brain tumor segmentation and organ segmentation tasks, as well as classification tasks for colorectal cancer.

- **BTCV (Brain Tumor Segmentation Challenge 2015):** The BTCV dataset consists of multi-modal MRI scans of brain tumor patients, including T1, T1c, T2, and FLAIR modalities. The dataset contains 30 training and 10 test subjects, each with pixel-level annotations for the Whole Tumor (WT), Tumor Core (TC), and Enhancing Tumor (ET) regions. This dataset presents a challenging segmentation task due to the variability in tumor shape, size, and location across different patients.

- **MSD (Medical Segmentation Decathlon):** The MSD challenge provides a collection of 10 diverse datasets for multi-organ segmentation, including but not limited to the Spleen, Liver, Kidneys, and Prostate. Each dataset is annotated with pixel-wise labels. In particular, the Spleen dataset and the Multi-Modal Whole Heart Segmentation (MM-WHS) dataset are employed in our experiments. The MSD challenge is highly regarded for its diversity in terms of organs, imaging modalities (CT and MRI), and data size, making it a robust benchmark for evaluating segmentation algorithms.

- **BraTS 21 (Brain Tumor Segmentation 2021):** The BraTS 21 dataset is a collection of multi-modal MRI scans focused on brain tumor segmentation. It contains 1251 training cases and 438 test cases, each with multi-channel MRI images (T1, T1c, T2, and FLAIR), along with annotations for the Whole Tumor (WT), Tumor Core (TC), and Enhancing Tumor (ET). This dataset provides a comprehensive evaluation environment for segmentation methods, with substantial challenges due to the presence of varied tumor shapes, sizes, and complex brain anatomy.

- **CC-CCII (Colorectal Cancer Classification):** The CC-CCII dataset is a colorectal cancer classification dataset comprising CT scans of colorectal regions with pixel-level annotations indicating cancerous and non-cancerous regions. This dataset offers a different challenge, focusing on classification tasks and the need to distinguish between subtle variations in tissue structures and abnormal regions in CT images.

## A.2 Baselines

We compare the performance of STC3D against a diverse set of baseline models, including general self-supervised learning (SSL) methods and models specifically designed for medical image analysis. These baselines cover various approaches to image segmentation and classification, utilizing both traditional and state-of-the-art methods in self-supervised learning.

- **UNETR:** UNETR is a transformer-based model designed for 3D medical image segmentation. It integrates a Vision Transformer (ViT) backbone within the U-Net architecture, which allows it to capture long-range dependencies in 3D volumes. This model has been shown to outperform conventional CNN-based methods in capturing contextual information.

- **Swin-UNETR:** An extension of UNETR, Swin-UNETR incorporates the Swin Transformer, a hierarchical vision transformer that excels at capturing multi-scale features. Swin-UNETR has demonstrated superior performance in multi-scale medical image segmentation tasks due to its ability to capture both local and global image patterns.

- **MAE3D (Masked Autoencoder 3D):** MAE3D is a self-supervised method for 3D medical images, leveraging a masked autoencoder framework. This method trains the model by masking portions of the input volume and learning to reconstruct the missing regions. It is known for its effectiveness in pre-training large models using unlabelled data.

- **SimCLR:** SimCLR is a popular contrastive learning framework where the model learns to distinguish between positive and negative image pairs by maximizing the similarity of positive pairs and minimizing the similarity of negative pairs. This approach relies on data augmentation to generate positive pairs.

- **SimMIM (Masked Image Modeling):** SimMIM applies a self-supervised learning approach where masked regions of the input image are used to learn contextual representations. This method has been successful in both image classification and segmentation tasks.

- **MoCo v3 (Momentum Contrast):** MoCo v3 is an enhanced contrastive learning approach that maintains a dynamic memory bank (queue) of representations to enable the learning of high-quality image embeddings. It has been shown to be effective in large-scale unsupervised learning tasks.

- **Jigsaw:** The Jigsaw method is based on a self-supervised task that divides images into patches, which are then rearranged and the model is trained to predict the correct patch order. This approach encourages the model to learn spatial relationships between different parts of the image.

- **PositionLabel:** PositionLabel is a self-supervised method that utilizes positional labels to aid in the pre-training process. The model learns to predict the location of patches within the image, which helps in learning robust image representations.

- **MG, ROT, Rubik++, PCRLv1:** These are medical-specific SSL methods. MG (Multi-Granularity) focuses on multi-scale and multi-level features for segmentation. ROT leverages rotation-based augmentation for self-supervised learning. Rubik++ uses a Rubik's cube-like puzzle to train the model to solve spatial challenges. PCRLv1 (Position-Contrastive Regularization) applies position-based regularization during contrastive learning for improved robustness in medical image analysis tasks.

### A.3 STC3D Model Parameters

In this subsection, we detail the parameter settings used for the STC3D model in our experiments. The settings are categorized into pre-processing, pre-training, and fine-tuning stages, and are summarized in subsections.

#### A.3.1 Pre-processing Settings

The pre-processing stage involves preparing the input data for training. The parameters for this stage are as follows:

- **Spacing:** The spacing between voxels is set to $[1.5, 1.5, 1.5]$, which determines the voxel resolution in the three-dimensional space.

- **Normalization:** The intensity values of the input volumes are normalized to the range $[-175.0, 250.0]$ for $[a_{\min}, a_{\max}]$ and $[0.0, 1.0]$ for $[b_{\min}, b_{\max}]$, ensuring that the voxel intensities are scaled appropriately for neural network processing.

- **ROI Size:** The region of interest (ROI) is set to a size of $64 \times 64 \times 64$, which determines the size of the input patch during training.

- **Augmentation:** Random rotations and flips are applied to the input data to augment the dataset and improve generalization during training.

#### A.3.2 Pre-training Settings

The pre-training stage involves training the model in a self-supervised manner to learn useful feature representations. The pre-training settings include:

- **Pre-training steps:** The model is pre-trained for $100k$ steps.

- **Optimizer:** The AdamW optimizer is used with a learning rate of $1 \times 10^{-3}$.

- **Learning Rate Schedule:** A warm-up cosine learning rate schedule is applied, with a warm-up period of 100 steps.

- **Momentum:** Momentum is set to 0.9, which helps accelerate convergence by smoothing gradients.

- **Regularization weight:** The regularization weight is set to $1 \times 10^{-2}$ to control overfitting.

- **Batch size:** The batch size is set to 4, which refers to the number of samples processed in each training step.

- **Sw batch size:** The sliding window batch size is also set to 4.

#### A.3.3 Fine-tuning Settings

The fine-tuning stage involves adapting the pre-trained model to the specific downstream tasks, such as segmentation or classification. The fine-tuning settings are as follows:

- **Optimizer:** The AdamW optimizer is used, with a learning rate of $3 \times 10^{-4}$.

- **Learning Rate Schedule:** A warm-up cosine learning rate schedule is also used in the fine-tuning phase, with a warm-up period of 100 steps.

- **Momentum:** Momentum remains set to 0.9 to maintain consistency in training behavior.

- **Regularization weight:** The regularization weight for fine-tuning is set to $1 \times 10^{-5}$.

- **Batch size:** The batch size for fine-tuning is set to 1.

- **Sw batch size:** The sliding window batch size for fine-tuning is also set to 4.

- **Inference:** A sliding window approach is used during inference to process large volumes.

- **ROI Size:** The ROI size for fine-tuning is set to $96 \times 96 \times 96$, which increases the resolution compared to the pre-training stage.

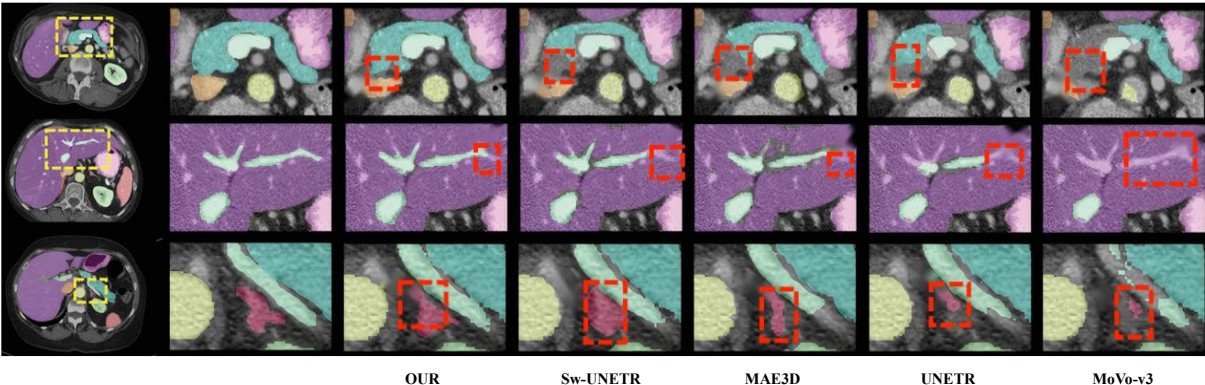

Figure 4: Comparison of segmentation results from STC3D and other state-of-the-art models. The first column shows the input CT image with the corresponding ground truth segmentation in color. The subsequent columns display the segmentation outputs from the following models: our STC3D, Swin-UNETR, MAE3D, UNETR, and MoVo-v3. The red boxes highlight areas where our model achieves more accurate segmentation, particularly in challenging regions with complex anatomical structures. The yellow boxes indicate the regions of interest in the original input image for closer inspection of model performance.

## A.4 Invariant Representation Learning

Invariant Representation Learning aims to learn features that are invariant to transformations applied to the data. The goal is to ensure that the model learns representations that are robust to changes in spatial transformations such as rotation, translation, and scaling, without losing the essential structure of the data.

In mathematical terms, this concept can be formalized by learning a mapping $f : X \to \mathcal{Z}$, where $X$ represents the input space (e.g., images) and $\mathcal{Z}$ represents the feature space. We aim for the following properties:

$$\mathbb{E}_{T \sim \mathcal{T}} \left[ d(f(x), f(T(x))) \right] \leq \epsilon \quad \forall x \in X, \forall T \in \mathcal{T} \tag{15}$$

Where: - $T(x)$ is a transformed version of $x$ under a transformation $T \in \mathcal{T}$ (e.g., rotation, translation). - $d(\cdot, \cdot)$ is a distance metric in the feature space, often chosen as Euclidean distance or cosine similarity. - $\epsilon$ is a small error tolerance.

This formulation ensures that the learned representation $f(x)$ is invariant to the transformations in $\mathcal{T}$. A commonly used approach to achieve this is contrastive learning, where positive pairs of transformed samples are encouraged to have similar representations.

## A.5 Regularized Learning

Regularized Learning introduces additional constraints into the optimization process to prevent overfitting and encourage the model to learn more robust and generalized features. The idea is to penalize overly complex models by adding a regularization term to the objective function.

Consider the objective function in a supervised learning setting:

$$L_{\text{total}} = L_{\text{loss}}(f(x), y) + \lambda R(f) \tag{16}$$

Where: - $L_{\text{loss}}(f(x), y)$ is the loss function (e.g., cross-entropy or mean squared error) that measures the discrepancy between the predicted output $f(x)$ and the true label $y$. - $R(f)$ is the regularization term, which could be $\ell_1$ or $\ell_2$ regularization or other forms of regularizers that control the complexity of the learned model. - $\lambda$ is the regularization hyperparameter that controls the strength of the regularization.

For instance, the $\ell_2$-regularized learning has the form:

$$R(f) = \sum_i \|\theta_i\|_2^2 \tag{17}$$

Where $\theta_i$ represents the model parameters. The regularization term encourages the model to minimize the magnitude of the parameters, preventing it from overfitting to the training data and ensuring better generalization to unseen data.

In the case of contrastive learning, regularization can be implemented by enforcing distinct representations for different regions of the input space. This can be achieved through a regularization loss term $L_{\text{reg}}$ that ensures the feature discrepancy between different anatomical regions is maximized.

$$L_{\text{reg}} = \sum_{i,j} \|f(x_i) - f(x_j)\|_2^2 \tag{18}$$

Where $x_i$ and $x_j$ are two different views or augmentations of the input, and $f(x)$ is the feature representation learned by the model.

## A.6 Domain-Invariant Learning

Domain-Invariant Learning is the process of learning representations that are generalizable across different domains, such as datasets with different imaging conditions, resolution, or anatomical structures. The key idea is to learn features that are not specific to any one domain but instead capture the underlying shared structure across multiple domains.

This concept can be modeled using adversarial learning or domain adaptation techniques. A common approach in domain-invariant learning is to use a domain classifier $D$ to predict the domain from which a sample $x$ originates, while simultaneously training a feature extractor $f$ to fool this classifier.

The total loss for the domain-invariant learning setup can be written as:

$$L_{\text{total}} = L_{\text{task}}(f(x), y) - \lambda L_{\text{domain}}(f(x), D(x)) \tag{19}$$

Where: - $L_{\text{task}}(f(x), y)$ is the task-specific loss (e.g., segmentation or classification). - $L_{\text{domain}}(f(x), D(x))$ is the domain classification loss, where $D(x)$ predicts the domain of $x$ based on its feature representation $f(x)$. - $\lambda$ is the hyperparameter that controls the trade-off between task performance and domain invariance.

The objective of domain-invariant learning is to minimize the domain loss $L_{\text{domain}}$, which ensures that the model cannot distinguish between samples from different domains, thus learning domain-invariant features. This can be achieved through techniques like adversarial training, where the model learns to extract features that are not discriminative for domain classification.

In mathematical terms, domain-invariant features are learned by minimizing the following objective:

$$L_{\text{domain-invariant}} = \mathbb{E}_{x \sim X}\left[\|f(x) - \bar{f}(x)\|_2^2\right] \tag{20}$$

Where $\bar{f}(x)$ is the target domain feature representation (possibly from a different domain or dataset), and $\|\cdot\|_2^2$ represents the squared Euclidean distance between the features from different domains.

This formulation encourages the model to learn representations that are invariant across domains while still being useful for the primary task at hand.

### A.7 Performance comparison with different cropping strategies

Table 7: Performance comparison on the BTCV dataset with different cropping strategies.

| Cropping Strategy | Without Regularization (Dice Score) | With Regularization (Dice Score) | Without Regularization (IoU) | With Regularization (IoU) |
|---|---|---|---|---|
| Random Cropping | 77.85 ± 1.1 | 82.01 ± 0.8 | 64.35 ± 1.2 | 68.54 ± 1.0 |
| Center Cropping | 80.12 ± 0.8 | 85.34 ± 0.7 | 67.23 ± 1.0 | 71.12 ± 0.9 |
| Edge Cropping | 75.23 ± 1.2 | 80.45 ± 0.9 | 62.11 ± 1.4 | 66.92 ± 1.0 |

In the experiments on the BTCV dataset, regularization branch significantly improved the performance of all cropping strategies. Random cropping performed in between the other two strategies, with a Dice score of 77.85% without regularization, improving to 82.01% with regularization. Center cropping performed the best without regularization (Dice score of 80.12%) and further improved to 85.34% with regularization. Edge cropping performed the worst, with a Dice score of 75.23% without regularization, improving to 80.45% with regularization. The regularization branch played a key role in enhancing feature consistency and improving the model's robustness to anatomical variations, especially in preventing feature repulsion caused by different cropping strategies.

### A.8 Novelty and Handling of Anatomical Variations

The proposed STC3D method demonstrates innovation in addressing anatomical variations in 3D medical images. While many studies have explored augmentation strategies to improve self-supervised contrastive learning, STC3D improves upon this by introducing controlled spatial transformations, such as rotation, translation, and scaling. These transformations not only simulate natural anatomical variations but also enable the model to learn robust representations that remain consistent across various structural changes. Unlike traditional methods that rely on simple augmentations like random cropping, controlled transformations help preserve the spatial relationships within anatomical structures, ensuring that the model can adapt to variations observed in different patients.

In conventional contrastive learning frameworks such as SimCLR and MoCo, random cropping and simple augmentation methods (e.g., color jittering, cropping) are widely used. However, these methods fail to adequately account for the complex anatomical variations across patients. For instance, in brain tumor segmentation, different patients have tumors of varying sizes, positions, and shapes. Random cropping may result in losing critical anatomical information, especially when it cuts out important structures. In contrast, STC3D, by utilizing controlled transformations such as rotation, translation, and scaling, simulates more realistic anatomical changes, such as different orientations or sizes of organs, which are commonly encountered in medical imaging. These transformations allow the model to better generalize across patients, making it more robust to anatomical variations and imaging conditions. STC3D's novelty is also reflected in the introduction of a regularization branch. This branch enhances feature learning by maximizing the discrepancy between features from different slices or views, encouraging the model to be sensitive to fine anatomical differences. The regularization branch helps prevent overfitting to specific anatomical regions caused by transformations like cropping, and thus improves the model's ability to adapt to various anatomical structures.

### A.9 High-Level Semantic Learning in Contrastive Frameworks

While traditional self-supervised learning (SSL) methods, including SimCLR and MoCo, often fail to capture high-level semantics during pre-training, the introduction of controlled spatial transformations in STC3D enables a more effective learning of these semantics. SimCLR and MoCo, which are typically applied to 2D image data, rely on simple augmentations (e.g., random cropping, color jittering) to learn contrastive representations. However, such augmentations primarily focus on low-level features and often overlook the

complex spatial relationships inherent in medical images. In contrast, STC3D applies controlled transformations such as rotation, translation, and scaling to 3D medical volumes, creating multiple views that simulate real-world anatomical variations. This allows the model to learn more meaningful, context-aware representations that go beyond low-level features and capture the high-level semantics of anatomical structures. By leveraging these spatial transformations, STC3D can better understand the relationships between different parts of the anatomy, facilitating the model's ability to generalize to unseen data.

Moreover, the addition of a regularization branch in STC3D further contributes to the learning of high-level semantics by promoting feature diversity. Unlike traditional contrastive learning methods that only minimize the distance between positive pairs and maximize it between negative pairs, the regularization branch in STC3D forces the model to learn more discriminative features between different slices or views. This approach improves the model's ability to distinguish between subtle anatomical structures, thus enabling the model to capture more complex, high-level semantic information. The regularization ensures that the learned representations maintain meaningful semantic relationships across different views, even as anatomical features undergo various transformations. This shift from simple augmentations to more structured transformations is key to STC3D's ability to learn high-level semantics effectively, especially in the context of 3D medical image analysis.

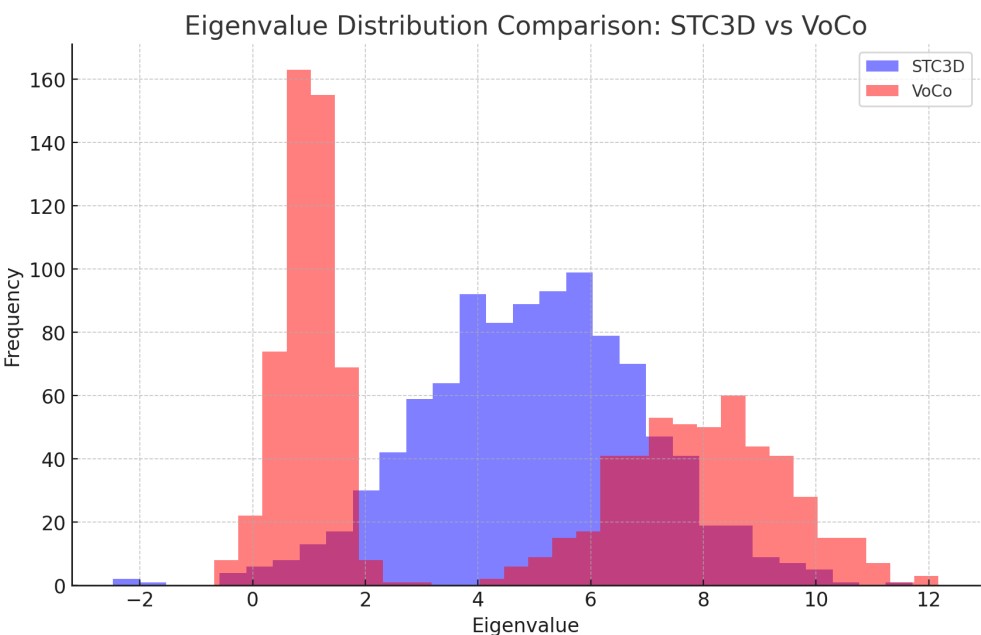

Figure 5: Eigenvalue Distribution Comparison: STC3D vs VoCo. The histogram shows the distribution of eigenvalues for both models. STC3D exhibits a more spread-out distribution, indicating a broader feature space, while VoCo shows a more concentrated distribution, suggesting dimensional collapse.

### A.10 Regularization Branch and Feature Learning in STC3D

The regularization branch in STC3D plays a crucial role in enhancing feature learning by ensuring a broader feature space for different anatomical regions. Unlike traditional models like VoCo, which tend to suffer from dimensional collapse—where the learned features become overly compressed and fail to maintain distinct representations—STC3D encourages diversity in the feature space. This is achieved by promoting the discrepancy between features from different slices or views, effectively pushing the model to learn more discriminative and robust representations across different anatomical regions.

To demonstrate the impact of the regularization branch and its ability to prevent dimensional collapse, we analyze the eigenvalue distribution of the learned feature space. In the comparison between STC3D and VoCo, as shown in Figure 1, we observe that STC3D's feature values are more spread out across the feature

space, reflecting a more diversified feature representation. The histogram reveals that STC3D's eigenvalues follow a more normal distribution, indicating that the model maintains a rich set of features without excessive compression. In contrast, VoCo shows a more concentrated distribution of eigenvalues, particularly around a few specific values, suggesting that the model is experiencing dimensional collapse. This concentration of eigenvalues in VoCo points to a loss of diversity in the feature space, where certain anatomical structures may become indistinguishable due to overfitting to a narrow set of features.(From Fig.5)

