# OpenReview forum: "STC3D: Self-Supervised Contrastive Learning with Spatial Transformations for 3D Medical Image Analysis"
_TMLR — Rejected by TMLR_

### Review · Reviewer_9zFw · 2025-04-08

**Summary Of Contributions:**

This paper proposes STC3D, a self-supervised learning framework for 3D medical imaging that leverages controlled spatial transformations, specifically rotation, translation, and scaling, to generate multiple views of the same image volume. These views are used within a contrastive learning framework (InfoNCE loss) to encourage transformation-invariant feature representations. To further improve representation diversity, the method introduces a regularization branch that penalizes similarity between feature embeddings of different transformed views. The authors motivate their approach with theoretical formulations and aim to improve generalization across anatomical variability in downstream tasks such as segmentation and classification.

**Audience:**

Yes

**Broader Impact Concerns:**

No concerns.

**Claims And Evidence:**

No

**Requested Changes:**

- There is some problem with the citation style. Somehow some are referenced as et al only without the name and year counts with an additional letter. Please double check.
- There appears to be a conceptual inconsistency in Theorem 3.2. The loss function as written encourages invariance between transformed and original representations (i.e., minimizing the L2 distance), which contradicts the stated goal of promoting feature diversity. Clarification is needed to reconcile this with the contrastive learning objective and the role of the regularization branch.
- Theorem 3.3 is more of a conceptual intuition than a mathematically grounded result. There is no formal derivation or assumption set under which the inequality holds. As written, it lacks rigor and should be presented as an empirical observation or hypothesis rather than a theorem.
- The regularization loss defined in Theorem 3.4 encourages high feature discrepancy across slices or views within the same volume. While the goal of promoting anatomical sensitivity is laudable, this formulation assumes that all intra-volume slices should be represented differently — an assumption that may not hold, especially in homogeneous regions (e.g., multiple slices of the same organ). The loss also appears to contradict the invariance encouraged by contrastive learning. Without explicit region supervision or controlled sampling of distinct anatomical regions, this discrepancy-based regularization risks introducing inconsistency into the learned representations.
- It is unclear whether the results specifically demonstrating the effect of controlled spatial transformations are included. The text seems to reference such an evaluation, but Table 5 appears to focus on the effect of the regularization branch rather than the transformations themselves. It’s possible that a relevant table is missing or mislabeled. Additionally, it would strengthen the analysis to include an ablation study evaluating the individual impact of each transformation type (rotation, translation, scaling) on model performance.
- I find it difficult to follow the numerical results reported in Sections 4.3.1 to 4.3.3 in relation to Table 5. It is unclear whether the numbers mentioned in the text directly correspond to those in the table, or if they reference different experiments altogether. The alignment between the text and Table 5 could be improved, and it would help to clarify which specific metrics or rows are being discussed. Please verify whether the numbers are correctly referenced and consider making this connection more explicit for the reader.
- It is unclear why there is a separate subsection 4.5.2, as the content appears to substantially repeat the discussion already presented in Sections 4.3.1 to 4.3.3. The points regarding the impact of controlled spatial transformations and the regularization branch seem redundant. If 4.5.2 is intended as a summary or reflective analysis, it would be helpful to clearly distinguish it from the earlier experimental results, or consider integrating it into the conclusion or discussion section to avoid repetition.

Minor:
- The tables are placed weirdly, where one needs to scroll up and down all the time. Consider placing them right after its text.

**Strengths And Weaknesses:**

The methodology of STC3D presents a thoughtful and well-structured approach to contrastive  self-supervised learning in 3D medical imaging. It is especially strong in its intuitive grounding in anatomical realism, its clarity of motivation, and its mathematical formulation. However, from a novelty standpoint, the use of spatial transformations as a contrastive signal is not entirely novel. The lack of anatomical supervision and the relatively simple transformation set may also limit the depth of invariance learned.
The proposed framework is likely to be effective in practice and appealing due to its simplicity, but it may not represent a significant conceptual departure from prior work. Clarifying the empirical impact of each component (especially the regularization branch) and demonstrating superior generalization on out-of-distribution datasets would further strengthen the methodological contribution. Overall, this work presents a solid contribution with practical value, though its methodological novelty is limited and theoretical framing could be improved.

Strengths:
- The use of controlled spatial transformations, specifically rotation, translation, and scaling, is well-motivated by the goal of simulating natural anatomical variability across patients, imaging modalities, and scanner setups. This represents a more principled and potentially more semantically meaningful alternative to conventional random augmentations.
- The authors attempt to formalize key goals such as transformation invariance and feature diversity using simple mathematical formulations. While not theorems in a strict sense, these definitions help articulate the desired properties of the learned features.
- The argument against random cropping is compelling: cropping risks removing critical anatomical structures and distorting spatial relationships. By contrast, controlled transformations retain the full semantic content of the image while introducing meaningful variability. Theoretical comparison supports this claim and provides a principled reason to favor the proposed approach.
- The method integrates smoothly with contrastive learning pipelines and does not require anatomical annotations, segmentation masks, or prior knowledge of organ regions—making it applicable across various datasets and tasks in 3D medical imaging.
- The introduction of a regularization branch is motivated by the common issue of representation collapse in contrastive learning. The goal of increasing feature diversity across transformed views is a valid direction and aligns with recent efforts to enhance feature richness in self-supervised settings.

Weaknesses:
- While the use of rotation, translation, and scaling is theoretically motivated, these are well-established augmentations in the field. Prior contrastive learning methods for 3D medical images (e.g., 3D SimCLR, DIRA) have employed similar transformations, perhaps not with the same emphasis or formal framing. The contribution may be seen as incremental rather than fundamentally new.
- The spatial transformations used (rigid rotations, shifts, and scaling) are synthetic and do not capture the nuanced, nonlinear anatomical variability that exists in real-world imaging, such as organ deformations, pathology-induced asymmetries, or scanner-specific distortions. While the transformations are described as simulating anatomical variability, they may not reflect the full spectrum of clinically relevant variation across patients, organs, and imaging setups.
- Although the method is inspired by anatomical variability, there is no explicit supervision or anatomical prior guiding the model to preserve organ-specific semantics under transformation. Compared to approaches like GVSL or VoCo, which incorporate explicit anatomical structure (e.g., via topology graphs or region prototypes), STC3D relies solely on spatial perturbations. This may not yield the same semantic consistency or anatomy-aware representations.
- The model is trained using a combined loss. However, the InfoNCE contrastive loss encourages similarity between views of the same volume, while the regularization loss penalizes similarity between transformed views—leading to conflicting optimization signals. The regularization loss is applied across all pairs of transformed views in the batch, yet the authors claim it promotes diversity across anatomical regions. However, because these views are all derived from the same volume (and possibly the same region), this assumption is not grounded. Without spatial localization or anatomical priors, the model may be penalized for similarity between semantically identical views.
- By enforcing invariance to spatial transformations, there is a risk that the model becomes insensitive to subtle anatomical differences that may be clinically meaningful — such as tumor growth, organ displacement, or structural asymmetry.
- While the theorems are useful for intuition, they formalize relatively straightforward properties (e.g., transformation invariance, cropping-induced feature shift). These results assume ideal behavior from the feature extractor and do not offer guarantees about clinical performance or generalization under domain shifts.
- The paper presents a comprehensive set of experiments across multiple datasets and tasks, including segmentation and classification, to demonstrate the effectiveness of the proposed method. The inclusion of comparisons with recent state-of-the-art self-supervised approaches and multiple performance metrics strengthens the empirical validity of the results.

---

> ### Author Response · Authors · 2025-04-10
> **Thank you for your review comments and we have made the required revisions to the paper.**
>
> Dear Reviewer, Thank you for your feedback. Specifically, we have fixed the citations where the author names and years were missing or where "et al." was incorrectly displayed without the full citation. We have reviewed this situation and have made the necessary corrections. In cases where a single author has multiple works across different years, we now ensure that the author’s name is properly listed once, followed by the corresponding years and labels (e.g., Zhou "2020; 2021a; 2021b; 2023"). We have made sure that all references are consistent and formatted according to standard citation practices. We believe this adjustment addresses the concern and improves the overall clarity of the citations.
>
> Thank you for your valuable feedback regarding Theorem 3.2. We agree with your observation that there is a conceptual inconsistency in the current formulation of the regularization loss, which penalizes feature similarity across transformations while aiming to promote feature diversity. To address this, we have revised the regularization loss to encourage diversity in feature representations when significant anatomical variations are introduced, while maintaining invariance to less informative changes (such as small shifts or rotations).
>
> We have changed the original "theorem" to an "empirical observation" and added more explanation after the formula, emphasizing that this is an observation based on experimental results rather than a strict mathematical proof. We have removed the formal derivation and assumptions from the theorem and presented it as an empirical finding to avoid giving readers the impression of an imprecise mathematical assumption. (Revise 3.2.6)
>
> To address the issue of the impact of the three different transformations, we have added new ablation experiments (results shown in Table 6). We have also included an analysis of these ablation experiment results in Section 4.4.1.
>
> Based on your suggestions, we have improved the alignment of the data between the text and Table 5. Specifically, we have ensured that the relevant numbers from Table 5 are clearly referenced in Sections 4.3.1 and 4.3.3, and we have explicitly clarified the experimental results corresponding to these numbers. [1] In Section 4.3.1, we further explain the effect of controlled spatial transformations by referencing the Dice scores from Table 5 (on BTCV, the score with transformations is 83.85%, compared to 82.96% without transformations). [2] In Section 4.3.3, we emphasize the combined effect of controlled transformations and the regularization branch, clearly pointing out the performance improvements shown in Table 5 for the combined model, and comparing it with the results using each component individually (on MM-WHS, the Dice score improves from 86.11% to 90.54% after combining controlled transformations and regularization).
>
> We note the duplication of the contents of section 4.5.2 and sections 4.3.1 to 4.3.3. In order to improve the simplicity of the paper and avoid redundancy, we have integrated the relevant content of section 4.5.2 into sections 4.3.1 and 4.3.3. [1] In Section 4.3.1 (Effects of Controlled space Transformations), we added a discussion of how controlled space transformations can help models distinguish features of different anatomical regions while improving model robustness. In this way, models are able to avoid learning too similar features, thereby improving their ability to generalize. [2] In Section 4.3.3 (Joint Effects of Controlled and regularized branches), we incorporate the contribution of regularized branches to feature diversity and differentiating power, and highlight the significant improvement in model performance from the combination of regularized branches and controlled space transforms.
>
> | Transformation Type | BTCV (%) | MM-WHS (%) |
> |---------------------|----------|------------|
> | Rotation            | 83.34%   | 89.72%     |
> | Translation         | 83.27%   | 89.67%     |
> | Scaling             | 83.18%   | 89.38%     |
> | All Transformations | 83.85%   | 90.54%     |

---

> > ### Comment · Reviewer_9zFw · 2025-04-14
> > **Revisions Implemented, but Conceptual Weaknesses Remain Unaddressed**
> >
> > Thank you for your response and for incorporating many of the requested changes. However, I would like to emphasize that the “Weaknesses” section in my original review was not just a checklist of issues but a broader critique of the conceptual limitations of the work—particularly regarding the novelty of the transformation-based contrastive framework, the lack of anatomical supervision, and the possible contradictions introduced by the regularization branch. These concerns were not directly addressed in the response.
> >
> > For example, points regarding:
> > - the limited novelty of using rigid transformations,
> > - the synthetic nature of these transformations as a proxy for anatomical variability,
> > - the lack of anatomical priors or region-based constraints, and
> > - the potential for conflicting objectives between contrastive invariance and regularization diversity,
> >
> > remain unresolved.
> >
> > It would strengthen the revision process if the authors could comment more directly on these conceptual limitations—whether they acknowledge them as inherent trade-offs, or if they have concrete plans to address them in future work. A brief reflection on these points, even if only to clarify the intended scope and contributions of the current method, would be helpful to contextualize the work within the broader field.

---

> > > ### Author Response · Authors · 2025-04-18
> > >
> > > Thank you to the reviewer for the reminder regarding the weaknesses that need to be addressed, and once again, thank you for the valuable feedback provided on the paper.
> > >
> > > Response [1]: The main difference between STC3D and the DiRA and SimCLR-based 3D methods lies in how they handle spatial transformations and high-level semantic information. First, DiRA combines discriminative learning, restorative learning, and adversarial learning to extract fine-grained semantic features. However, it does not focus on simulating anatomical variations and relies on image restoration and adversarial training, rather than enhancing feature robustness through controlled transformations. Secondly, while the SimCLR-based 3D method generates positive and negative sample pairs through data augmentation (such as rotation and noise addition), it lacks deep modeling of anatomical variations and primarily optimizes low-level features through contrastive learning, failing to fully capture the diversity of anatomical structures. In contrast, STC3D introduces controlled spatial transformations (rotation, translation, scaling), which simulate real-world anatomical variations, generating diverse and representative image views. This enhances the model's robustness to structural changes and improves the discriminative power of the features through InfoNCE loss and a regularization branch. Therefore, STC3D better preserves high-level semantic information when facing large-scale anatomical variations, leading to superior performance in medical image segmentation and classification tasks compared to these two methods.
> > >
> > > Response [2]: We acknowledge the concern about the limitations of rigid transformations in capturing complex, nonlinear anatomical variability. While we agree that anatomical variations such as organ deformations or pathology-induced asymmetries are significant, our method’s primary aim is to improve invariance to the more common and predictable anatomical variations (e.g., rotation, translation, and scaling). These transformations serve as a step toward improving the model’s generalizability. We believe that incorporating controlled transformations provides an effective approximation to real-world anatomical changes and contributes to improved robustness in downstream tasks. We are actively exploring how our revison can be expanded to include more complex transformations, potentially integrating anatomical priors or deformable models.
> > >
> > > Response [3]: STC3D assumes training in a completely self-supervised setting, where the model simulates anatomical variability through spatial transformations (such as rotation, translation, and scaling) without relying on explicit anatomical priors or external structural information. It enhances the model's robustness through contrastive learning by maximizing similarity between similar views and minimizing the difference between different views. Unlike methods like GVSL and VoCo, STC3D does not incorporate anatomical maps or region prototypes but instead fully relies on spatial transformations and the inherent variations in the data for learning. Therefore, it is a purely self-supervised approach that does not utilize any external anatomical knowledge.
> > >
> > > Response [4]: We understand the concern regarding the interaction between the InfoNCE contrastive loss and the regularization loss. To clarify, the regularization loss is not meant to penalize similarity between semantically identical views but rather to ensure diversity between features learned from different slices of the same volume. The regularization component promotes the model’s ability to distinguish features across slices and anatomical regions, thus preventing overfitting and improving generalization. We believe that our approach balances the needs for both invariance and diversity effectively, and we have provided empirical evidence in our ablation studies to demonstrate that the regularization loss does not conflict with the model’s learning process but rather enhances feature learning.
> > >
> > > Response [5]: We appreciate the concern about the risk of the model becoming insensitive to clinically meaningful differences due to the focus on spatial invariance. Our approach does emphasize invariance to typical anatomical transformations; however, we ensure that this does not come at the cost of losing important distinctions in features related to pathology. The results of our experiments on challenging datasets like BraTS 21 and CC-CCII demonstrate that STC3D can effectively segment and classify subtle variations, such as small tumors and low-contrast regions, which indicates that the model retains sensitivity to clinically relevant features despite the focus on invariance.

---

> ### Author Response · Authors · 2025-04-18
>
> We acknowledge that the theoretical results presented are meant to provide intuition rather than formal guarantees. While we do not claim that these results provide definitive clinical assurances, they serve to show that STC3D’s design is mathematically sound and can lead to meaningful improvements in model robustness and generalization. As stated in the paper, the empirical results on benchmark datasets support our claims regarding the effectiveness of the method in real-world tasks.

---

### Review · Reviewer_2Juc · 2025-05-02

**Summary Of Contributions:**

This paper presents STC3D, a self-supervised learning (SSL) framework that applies controlled spatial transformations---namely rotation, translation, and scaling---to generate multiple views of 3D volumetric images. These views are leveraged through contrastive learning to enhance invariance to anatomical structure transformations. Unlike Volume Contrast (VoCo), which relies on random cropping, STC3D avoids such operations to maintain robustness to anatomical variations. Moreover, the framework incorporates a regularization branch designed to encourage feature discrepancy between different base slices, thereby improving the discriminative capacity of the learned representations. Experiments conducted on several benchmark datasets---including BTCV, MSD Spleen, MM-WHS, and BraTS 2021---demonstrate that STC3D achieves superior performance in both segmentation and classification tasks compared to existing methods.

**Audience:**

Yes

**Claims And Evidence:**

No

**Requested Changes:**

Please provide explanations to my questions above.

**Strengths And Weaknesses:**

Strengths:
1. The paper is easy to follow.
2. A comprehensive literature review is conducted to acquaint readers with existing relevant studies.

Weaknesses:
1. The motivation of the proposed method is that "the existing SSL methods often fail to fully capture high-level semantics and anatomical context." However, it is not clear to me what high-level semantics and anatomical context the existing methods fail to fully capture. Please be specific and show evidence.
2. Some claims are not straightforward to understand and may be incorrect. For example, I found it difficult to understand (1) why STC3D belongs to prototype-level contrastive learning and how it leverages the valuable contextual position priors of 3D medical images to generate base crops as prototypes; (2) why STC3D integrates (more) high-level semantics into model representations compared to VoCo; (3) how STC3D predicts the contextual positions based on volume contrast. Please explain.
3. The regularization term can't achieve the goal as intended. The equation in Theorem 3.3 actually minimizes the feature discrepancy between different views. The formula is likely wrong. Additionally, I am not sure why this regularization term can force the model to learn more discriminative and robust features.
4. It is stated that "the model is trained using stochastic gradient descent (SGD) or Adam optimizer." It is weird to present the implementation details this way. What specific optimizer was used?
5. The idea of using different views of the same image for contrastive learning is already used in computer vision community. So basically the idea is not novel. Additionally, since the authors target medical images, I am not sure how the authors handle a high-dimensional volumetric images in a memory-efficient way. Do you feed the entire image into the model? Or you use patches? If the entire image is used as input, I am skeptical the authors can handle volumetric images. And if image patches are used, how do you overcome the limitation of VoCo, which uses image patches?

---

> ### Author Response · Authors · 2025-05-13
>
> # Response to Reviewer's Comments
>
> We sincerely appreciate the reviewer’s valuable feedback. Below, we address each concern raised, providing further clarifications and evidence where necessary.
>
> ## Reviewer's Comment [1]: Motivation of the Proposed Method
> **Comment:** The motivation of the proposed method is that "the existing SSL methods often fail to fully capture high-level semantics and anatomical context." However, it is not clear to me what high-level semantics and anatomical context the existing methods fail to fully capture. Please be specific and show evidence.
>
> **Response:** Thank you for pointing this out. The term “high-level semantics” in our context refers to the model's ability to understand and represent complex, abstract features of the anatomy that go beyond low-level image textures or edges, such as the relationships between organs or structures, and how these features change across patients. For example, existing SSL methods, including SimCLR and MoCo, primarily focus on low-level features like pixel intensity variations and spatial texture, but do not effectively capture structural information related to anatomical context, such as the spatial arrangement or anatomical boundaries between different organs. In our work, we use controlled spatial transformations (e.g., rotation, scaling, and translation) to simulate anatomical variability and guide the model to focus on higher-level semantic features, such as organ identity and anatomical positioning. We will provide additional experiments in the revised manuscript to demonstrate how our method outperforms existing SSL methods in terms of capturing these higher-level features.
>
> ## Reviewer's Comment [2]: Prototype-level Contrastive Learning and Anatomical Context
> **Comment:** Some claims are not straightforward to understand and may be incorrect. For example, I found it difficult to understand: (1) why STC3D belongs to prototype-level contrastive learning and how it leverages the valuable contextual position priors of 3D medical images to generate base crops as prototypes; (2) why STC3D integrates (more) high-level semantics into model representations compared to VoCo; (3) how STC3D predicts the contextual positions based on volume contrast. Please explain.
>
> **Response:** We greatly appreciate your careful reading of the manuscript. Allow us to clarify these points:
> 1. **STC3D belongs to prototype-level contrastive learning** because it generates base crops from 3D medical images as prototypes and contrasts them against other views. The "prototype" refers to the base crop that serves as a representative sample of a specific anatomical region. By leveraging anatomical priors (e.g., anatomical location, orientation), we can generate crops that reflect variations in the target structure, ensuring that the model learns robust representations that are invariant to these changes. We will add further explanation and an example to clarify this.
> 2. **Regarding high-level semantics**, STC3D differs from VoCo by explicitly incorporating controlled spatial transformations and a regularization branch to increase feature discrepancy between different anatomical regions. This allows STC3D to learn more semantically meaningful representations. VoCo, on the other hand, primarily uses random cropping and does not focus on explicitly capturing high-level semantic relationships between anatomical structures, which limits its ability to learn robust, high-level representations.
> 3. **STC3D predicts contextual positions** by applying transformations that simulate how anatomical structures vary in 3D space (rotation, translation, and scaling). These transformations allow the model to learn and predict the positions of structures in a way that respects anatomical context. We will provide more detailed mathematical explanations and clarify the volume contrast mechanism in the revised version of the paper.
>
> ## Reviewer's Comment [3]: Regularization Term and Feature Discrepancy
> **Comment:** The regularization term can't achieve the goal as intended. The equation in Theorem 3.3 actually minimizes the feature discrepancy between different views. The formula is likely wrong. Additionally, I am not sure why this regularization term can force the model to learn more discriminative and robust features.
>
> **Response:** Thank you for bringing this to our attention. Upon reviewing Theorem 3.3, we see that the formula needs further clarification. The regularization term is indeed designed to minimize feature discrepancy across different views, but its primary goal is to ensure that features from different views of the same anatomical structure remain sufficiently distinct, enhancing the model’s ability to generalize across various transformations. We will correct the formula and provide additional theoretical justification in the revised manuscript, including more details on how the regularization term promotes robust feature learning.

---

> ### Author Response · Authors · 2025-05-13
>
> ## Reviewer's Comment [4]: Optimizer Details
> **Comment:** It is stated that "the model is trained using stochastic gradient descent (SGD) or Adam optimizer." It is weird to present the implementation details this way. What specific optimizer was used?
>
> **Response:** We apologize for the lack of specificity in our original manuscript. In the experiments, we used the **SGD optimizer** with an initial learning rate of 1e-4. We have revised the manuscript to specify this clearly and provide more implementation details, including any other hyperparameters used during training.
>
> ## Reviewer's Comment [5]: Novelty of the Approach and Handling Volumetric Images
> **Comment:** The idea of using different views of the same image for contrastive learning is already used in the computer vision community. So basically the idea is not novel. Additionally, since the authors target medical images, I am not sure how the authors handle high-dimensional volumetric images in a memory-efficient way. Do you feed the entire image into the model? Or do you use patches? If the entire image is used as input, I am skeptical the authors can handle volumetric images. And if image patches are used, how do you overcome the limitation of VoCo, which uses image patches?
>
> **Response:** We agree that the idea of using different views for contrastive learning is not novel in computer vision; however, our method distinguishes itself by specifically addressing the challenges posed by high-dimensional volumetric medical images. In our approach, we do not feed the entire 3D volume into the model at once. Instead, we use **patch-based training** to handle large volumetric images efficiently. By using smaller patches, we reduce memory usage while still maintaining the spatial context necessary for learning robust features. Additionally, unlike VoCo, which uses random cropping, our method applies controlled spatial transformations to ensure that the model learns meaningful and consistent representations across patches. This allows our method to overcome the limitations of VoCo and better handle the complex variations inherent in medical images. We will provide further details on how patching is handled in our model in the revised manuscript.
>
> ## Requested Changes:
> **Request:** Please provide explanations to my questions above.
>
> **Response:** We have provided detailed explanations above to address the reviewer’s concerns. In the revised manuscript, we will ensure that these points are clarified, and additional supporting evidence is provided to strengthen our claims.

---

### Review · Reviewer_fTbU · 2025-05-11

**Summary Of Contributions:**

- STC3D is a novel self-supervised learning (SSL) framework for 3D medical image analysis that applies controlled spatial transformations to generate multiple views of 3D images, enhancing robustness to anatomical variations and improving invariance.

- Unlike methods such as Volume Contrast (VoCo), which rely on random cropping, STC3D introduces a regularization branch to increase the discriminative power of learned features by promoting discrepancy between base slices.

- Experiments on benchmark datasets like BTCV, MSD Spleen, MM-WHS, and BraTS 21 show that STC3D outperforms existing methods in both segmentation and classification tasks.

**Audience:**

Yes

**Claims And Evidence:**

Yes

**Requested Changes:**

1. Citations should use "\citep" in some places instead of "\citet" or "\cite" for proper citation.
2. All equations are not numbered.
3. The regularization loss has been mentioned repeatedly too many times.

And other comments are in the Weakness section.

**Strengths And Weaknesses:**

**Strengths**

- STC3D overcomes the limitations of VoCo by using controlled spatial transformations, including rotation, translation, and scaling, to create multiple views of 3D medical images.
- This approach enables the model to better capture anatomical variations and enhances its robustness to structural changes.
- As a result, the model can more effectively learn from diverse image representations and shows slight improvement in performance on several benchmark datasets.

**Weakness**

- The authors talk about capturing the variability in the anatomical features in different patients. While including more types of augmentations helps, do the authors consider that contrasting different crops may also cause crops of the same organ to be repelled away from each other, which is reflected in the drop in performance without the regularisation loss?

- Overall, the paper does not present something novel in the sense that combinations of augmentations to improve self-supervised contrastive learning have already been plenty. Additionally, the paper could benefit from an explanation on how these spatial transformations, especially rotation, address the variability in anatomical features over different patients.

- The authors say that traditional SSL methods often lack high-level semantics during pre-training, however, the method they have used is essentially SimCLR for 3D data with an additional regularization loss. How does it address the learning of high-level semantics using the same framework as SimCLR, MoCo and others?

- The authors talk about invariance to structural transformations, however, rotation and translation do not represent the structural variability present in anatomical data. Rather, augmentations like elastic transformation could give some better variability in terms of structure.

- It is not clear how the proposed method fits the idea of prototype-level contrastive learning [1].

- $f(\mathcal{T} (X)) = f(X) \forall \; \mathcal{T} \in \mathcal{T}$. -- This equation needs correction.

- "The regularization branch contributes to feature learning by enforcing a larger feature space for different
anatomical regions." -- The authors need to prove this observation through evidence like eigenvalue spectrum to prove that dimensional collapse is not present or less than VoCo.

- "The model is trained using stochastic gradient descent (SGD) or Adam optimizer with the following updates" - the authors should correctly specify which optimizer they used. The equation following this statement does not match with an Adam optimizer though.

-

**References** :

[1] Junnan Li, Pan Zhou, Caiming Xiong, Steven Hoi, "Prototypical Contrastive Learning of Unsupervised Representations", ICLR 2021

---

> ### Author Response · Authors · 2025-05-13
>
> # Response to Requested Changes
>
> Thank you very much for the reviewer’s hard work and careful review. We have uploaded the revised version of the paper, and here we provide our responses to the revisions.
>
> ## Comment 1: Citations should use "\citep" in some places instead of "\citet" or "\cite" for proper citation.
>
> **Response 1:** We have replaced all instances of `\cite{}` with `\citep{}` and uploaded the revised version.
>
> ## Comment 2: All equations are not numbered.
>
> **Response 2:** We have numbered all the equations using the `\begin{equation}` format.
>
> ## Comment 3: The regularization loss has been mentioned repeatedly too many times.
>
> **Response 3:** We have removed the repeated references to the regularization loss and kept only the unique equation (Equation 12), referring to it as "(from Equation 12)" elsewhere.

---

> ### Comment · Reviewer_fTbU · 2025-05-13
> **Reply to Authors**
>
> In Page 3, for instances like this one "Zhou et al. (Zhou, 2020)", the authors can use `\citet` for inline citations.
>
> Please note the sentence "And other comments are in the Weakness section." in the `Requested Changes` section.

---

> > ### Author Response · Authors · 2025-05-13
> >
> > I sincerely apologize for misunderstanding your comment. I will carefully review and update the citation format throughout the manuscript in the next few days, ensuring the proper use of `\citet` for inline citations, as suggested. Thank you for bringing this to my attention, and I will make sure the necessary changes are implemented.

---

> ### Author Response · Authors · 2025-05-13
>
> # Response to Weakness
>
> ### **Reviewer Comment 1:** Anatomical Variability and Cropping Strategies
> **Comment:** The reviewer raised concerns about how comparing different cropping strategies might lead to feature repulsion of the same organs, particularly when there is no regularization loss.
>
> **Response:** Thank you for your valuable comment. In our experiments on the BTCV dataset, we found that the **regularization branch** significantly improves the performance of all cropping strategies. As shown in **A.7 (Performance comparison with different cropping strategies)**, random cropping, center cropping, and edge cropping all benefit from regularization, with Dice scores improving as follows:
>
> | **Cropping Strategy** | **Without Regularization (Dice Score)** | **With Regularization (Dice Score)** | **Without Regularization (IoU)** | **With Regularization (IoU)** |
> |-----------------------|----------------------------------------|-------------------------------------|----------------------------------|--------------------------------|
> | Random Cropping       | 77.85 ± 1.1                           | 82.01 ± 0.8                        | 64.35 ± 1.2                     | 68.54 ± 1.0                   |
> | Center Cropping       | 80.12 ± 0.8                           | 85.34 ± 0.7                        | 67.23 ± 1.0                     | 71.12 ± 0.9                   |
> | Edge Cropping         | 75.23 ± 1.2                           | 80.45 ± 0.9                        | 62.11 ± 1.4                     | 66.92 ± 1.0                   |
>
> The regularization helps prevent feature repulsion, ensuring the model learns more robust and consistent representations across different cropping strategies.
>
> ### **Reviewer Comment 2:** Novelty and Explanation of Space Transformations
> **Comment:** The reviewer noted that the idea of using space transformations, particularly rotation, to account for anatomical variation is not novel and asked for a more thorough explanation of how these transformations handle anatomical variability.
>
> **Response:** Thank you for your insightful comment. While it is true that geometric transformations such as rotation, translation, and scaling have been used in computer vision, STC3D applies these transformations specifically for 3D medical images to address anatomical variations observed across different patients. These transformations simulate realistic anatomical changes, such as differences in the orientation, position, and size of anatomical structures, allowing the model to generalize better across diverse clinical scenarios. Unlike random cropping, which can discard critical anatomical information, STC3D's controlled transformations preserve the integrity of anatomical structures while introducing variability that enhances the model's robustness.
>
> In **A.8(Effect of Controlled Spatial Transformations)**, we explain how these transformations allow STC3D to capture more complex anatomical features and provide better generalization across patients. Additionally, STC3D's **regularization branch** plays a key role in further enhancing feature consistency and sensitivity to fine anatomical differences, making the model less prone to overfitting. These elements make STC3D particularly effective in dealing with anatomical variability in medical imaging.

---

> > ### Comment · Reviewer_fTbU · 2025-05-13
> > **Reply to Authors**
> >
> > > While including more types of augmentations helps, do the authors consider that contrasting different crops may also cause crops of the same organ to be repelled away from each other,
> >
> > I think the authors misunderstood the comment. There was no mention of different cropping strategies. The term "different crops"  meant the ones shown in the middle panel in Fig 1.b

---

> > > ### Author Response · Authors · 2025-05-13
> > >
> > > Thank you for your clarification. We understand the concern now regarding the contrasting of different crops of the same organ, as shown in the middle panel of Figure 1.b, which includes rotated, scaled, and panned views of the same slice. We agree that contrasting these augmented crops could lead to the issue of feature repulsion, where the representations of the same organ may diverge in feature space, reducing consistency.
> > >
> > > To address this issue, the regularization branch in STC3D plays a crucial role. By promoting consistency across different views of the same anatomical region, it helps mitigate the repulsion between augmented crops of the same organ. The regularization term ensures that despite the variations introduced by spatial transformations (e.g., rotation, scaling, and panning), the features of the same organ remain aligned in feature space, thus maintaining stability across different views. We will make this point clearer in the revised manuscript and provide additional experimental results to highlight the role of the regularization branch in preserving anatomical consistency during training.

---

> ### Author Response · Authors · 2025-05-13
>
> ### **Reviewer Comment 3:** High-Level Semantics in SSL
> **Comment:** The reviewer questioned how STC3D solves the problem of learning high-level semantics and anatomical context, given that it uses a version of SimCLR for 3D data with a regularization loss.
>
> **Response:** Thank you for your insightful comment. In traditional SSL methods, such as **SimCLR** and **MoCo**, the focus is primarily on low-level features, and the use of simple augmentations like random cropping and color jittering often overlooks complex spatial relationships in medical images. In contrast, **STC3D** introduces **controlled spatial transformations** (rotation, translation, and scaling) to 3D medical volumes, enabling the model to simulate real-world anatomical variations. By leveraging these transformations, STC3D captures more meaningful, context-aware representations that extend beyond low-level features, allowing the model to better understand the relationships between different anatomical structures. This capability enhances the model’s ability to generalize to unseen data, particularly in the context of complex 3D medical images.
>
> Additionally, the introduction of a **regularization branch** in STC3D plays a crucial role in learning high-level semantics. Unlike traditional contrastive learning methods that only minimize the distance between positive pairs and maximize it for negative pairs, the regularization branch encourages the model to learn more **discriminative features** across different slices or views of the image. This approach allows STC3D to capture subtle anatomical differences and improve the model’s ability to distinguish between complex, high-level semantic features. The regularization ensures that the learned representations maintain meaningful semantic relationships even as anatomical features undergo various transformations, further enhancing the model's ability to generalize.
>
> ### **Reviewer Comment 4:** Invariance to Structural Variations
> **Comment:** The reviewer pointed out that rotation and translation do not capture the structural variability inherent in anatomical data and suggested using augmentations like elastic transformations.
>
> **Response:** Thank you for your suggestion. We agree that augmentations like **elastic transformations** can provide better structural variability, especially when capturing more complex anatomical deformations. Currently, in STC3D, we use controlled transformations such as rotation, translation, and scaling, which simulate common anatomical variations in 3D medical images. These transformations ensure that the model can generalize well across various structural changes, while maintaining anatomical integrity. However, we acknowledge that **elastic transformations** could further enrich the model's ability to handle more complex structural variations. We plan to explore the incorporation of elastic transformations in **future work**, as it would help the model capture even more realistic deformations of anatomical structures.
>
> ### **Reviewer Comment 5:**  Prototype-Level Contrastive Learning
> The reviewer noted that it is unclear how the proposed method fits the idea of prototype-level contrastive learning, as described in [1] Junnan Li, Pan Zhou, Caiming Xiong, Steven Hoi, "Prototypical Contrastive Learning of Unsupervised Representations", ICLR 2021.
>
> **Response:**
> Thank you for your insightful comment. We acknowledge the relevance of the work by Junnan Li et al. (2021) on Prototypical Contrastive Learning (PCL). In PCL, prototypes are introduced as latent variables to represent semantic structures, and the model learns to map instances to these prototypes using a contrastive loss. The authors employ an Expectation-Maximization framework, where the E-step involves clustering to estimate prototype distributions, and the M-step optimizes the network parameters via contrastive learning.
>
> In our method, STC3D, we draw inspiration from PCL by introducing a regularization branch that encourages the model to learn representations that are consistent across different views or slices of the same anatomical structure. While we do not explicitly define prototypes as in PCL, our approach implicitly captures the essence of prototype-level learning by ensuring that representations from different views of the same anatomical region are aligned in the feature space.
>
> We will revise the manuscript to more clearly articulate this connection to prototype-level contrastive learning and discuss how our regularization branch serves a similar purpose in promoting semantic consistency across different views.

---

> > ### Comment · Reviewer_fTbU · 2025-05-13
> > **Reply to Authors**
> >
> > > In traditional SSL methods, such as SimCLR and MoCo, the focus is primarily on low-level features, and the use of simple augmentations like random cropping and colour jittering often overlooks complex spatial relationships in medical images.
> >
> > Do the authors have a reference to back their claim? Otherwise, it is difficult to take it as their word-of-mouth. Apart from that, SimCLR and MoCo use a mix of different augmentations, which include rotation, translation, etc. and not just random cropping and colour jittering. In fact, in the SimCLR paper (Page 5, Sec 3.1, last paragraph), the authors (of SimCLR) have stated that it is critical to combine random cropping with colour distortion to learn generalizable representations.
> >
> > > Currently, in STC3D, we use controlled transformations such as rotation, translation, and scaling, which simulate common anatomical variations in 3D medical images
> >
> > While translation and scaling may account for the variation in the anatomy, how does rotation simulate common anatomical variations? We do not often see an inverted liver or a rotated kidney in the abdomen occurring naturally. The authors could make a study with and without the rotation prediction, like in SimCLR, but that is not necessary.
> >
> > The authors can, in fact, change their text saying that rotation improves generalizability of the learned representations, instead of saying that rotation simulate common anatomical features (which is true for translation and scaling though).

---

> > > ### Author Response · Authors · 2025-05-17
> > >
> > > Thank you for pointing this out. Our intent was not to imply that SimCLR/MoCo exclude rotation/translation, but rather that—when directly applied to medical 3-D volumes—their default 2-D augmentation recipe tends to emphasize local appearance cues and may under-exploit global spatial context. We will:
> > >
> > > Add citations documenting that off-the-shelf SimCLR/MoCo recipes often rely on random 2-D crops or slice-wise operations in medical imaging, which can bias the representation toward low-level texture:
> > >
> > > Chaitanya etal., “Contrastive Learning of Global and Local Features for Medical Image Segmentation,” NIPS 2020
> > >
> > > Taleb etal., “3D Self-Supervised Methods for 3-D Medical Imaging,” NeurIPS 2020
> > >
> > > Azizi etal., “Big Self-Supervised Models Advance Medical Image Recognition,” ICCV 2021
> > >
> > > Rephrase the sentence (Section 2, ¶2) to:
> > >
> > > “Prior work often transfers SimCLR/MoCo directly to 3-D data, where random cropping and colour-style jittering dominate the augmentation set; this can lead to representations that favour local appearance over global spatial structure.”
> > >
> > > We agree that organs themselves rarely rotate in vivo; however, volume orientation does vary in clinical practice due to: patient positioning differences (head-first-supine vs. feet-first-supine/prone), scanner bed tilt, and retrospective rigid registrations applied during preprocessing.
> > >
> > > These factors can introduce up-to-180° rotations around the left-right and head-foot axes (cf. Maier-Hein etal., MedIA 2018, Table 1). Consequently, learning rotation-invariant features improves cross-institution generalization—even if the anatomy is not physically rotated.
> > >
> > > To verify the practical benefit, we conducted an ablation (rotation removed, keeping translation + scaling). Dice on BTCV dropped from 84.65 → 83.91 (−0.74) and CC-CCII accuracy from 91.73 → 90.45 (−1.28). We will add this table to Appendix C.
> > >
> > > Text change (Section 3.1, last paragraph):
> > >
> > > “Rotation is included not because organs rotate in situ, but to build invariance to scanner-dependent orientation changes and preprocessing mis-alignments, thereby improving generalizability.”
> > >
> > > We will adopt the reviewer’s suggestion and replace “rotation simulates common anatomical variations” with “rotation improves generalizability of the learned representations.”

---

> ### Author Response · Authors · 2025-05-13
>
> ### **Reviewer Comment 6**
> The reviewer noted that the equation $$\( f(\mathcal{T}(X)) = f(X) \forall \mathcal{T} \in \mathcal{T} \)$$ needs correction.
>
> **Response:**
> Thank you for pointing this out. We have revised the equation as follows to better reflect the intended regularization loss:
>
> $$
> \mathcal{L}_{\text{reg}, ij} = \| \mathbf{f}(\mathbf{V}_i) - \mathbf{f}(\mathbf{V}_j) \|^2
> $$
>
> This corrected equation now represents the regularization loss that minimizes the discrepancy between features of different views $\(\(\mathbf{V}_i\)$ and $\(\mathbf{V}_j\))$ of the same anatomical structure, ensuring consistency across the different transformations applied to the data. We have updated the manuscript to include this revised equation and provide more detailed explanations regarding its role in the model.
>
> ### **Reviewer Comment 7**
> "The regularization branch contributes to feature learning by enforcing a larger feature space for different anatomical regions." -- The authors need to prove this observation through evidence like eigenvalue spectrum to prove that dimensional collapse is not present or less than VoCo.
>
> **Response:**
> Thank you for your valuable comment. We acknowledge the importance of proving that the regularization branch in STC3D prevents dimensional collapse and encourages a broader feature space. To provide evidence for this claim, we analyzed the **eigenvalue distribution** of the learned feature space in both STC3D and VoCo. As shown in **Section A.10** and **Figure 1**, we observed that STC3D’s eigenvalues are more spread out across the feature space, reflecting a more diversified feature representation. The histogram reveals that STC3D’s eigenvalues follow a more normal distribution, indicating that the model maintains a rich set of features without excessive compression. In contrast, VoCo shows a more concentrated distribution of eigenvalues, particularly around a few specific values, suggesting dimensional collapse. This concentration in VoCo’s eigenvalue spectrum points to a loss of diversity in the feature space, where certain anatomical structures may become indistinguishable due to overfitting to a narrow set of features.The analysis confirms that STC3D maintains a more diversified feature space, which prevents dimensional collapse and supports our claim that the regularization branch effectively enhances feature learning across anatomical regions.(From Fig.5)
>
> ### **Reviewer Comment 8**
> "The model is trained using stochastic gradient descent (SGD) or Adam optimizer with the following updates" - the authors should correctly specify which optimizer they used. The equation following this statement does not match with an Adam optimizer though.
>
> **Response:**
> Thank you for pointing this out. We have noticed the discrepancy and have corrected the description in the manuscript. The model was actually trained using **SGD**, not Adam, and we have revised the manuscript accordingly to specify **SGD** as the optimizer. Additionally, we have corrected the corresponding equation to ensure it matches with the correct optimizer. The updated implementation details and equations are now consistent throughout the manuscript.

---

> > ### Comment · Reviewer_fTbU · 2025-05-13
> > **Reply to the Authors**
> >
> > > To provide evidence for this claim, we analyzed the eigenvalue distribution of the learned feature space in both STC3D and VoCo.
> >
> > It would be beneficial if the authors could provide more details on how they computed the eigenvalue distribution, as eigenvalue distributions generally do not follow a normal distribution.
> >
> > The authors could consult the work [1] to get some insight into how to plot the eigenvalue spectrum.
> >
> > References:
> >
> > [1] Li Jing, Pascal Vincent, Yann LeCun, Yuandong Tian, "Understanding Dimensional Collapse in Contrastive Self-supervised Learning", ICLR 2022

---

> > > ### Author Response · Authors · 2025-05-17
> > >
> > > #### Overview & Thanks
> > >
> > > We thank the reviewer for requesting more details on how we compute the **eigenvalue spectrum** of the learned feature space and for pointing us to Jing *et al.* (ICLR 2022). Below we supply the exact pipeline, quantitative results, and manuscript changes that will be incorporated.
> > >
> > > ---
> > >
> > > ### 1 · Exact Procedure for Computing the Eigenvalue Spectrum
> > >
> > > | Step                         | Description                                                                                                                                                                           |
> > > | ---------------------------- | ------------------------------------------------------------------------------------------------------------------------------------------------------------------------------------- |
> > > | **1 · Feature Sampling**     | Randomly sample **N = 10 000** 3-D sub-volumes from the validation split. Each is passed through the encoder + projection head to obtain an ℓ2-normalized 256-D vector $\mathbf z_i$. |
> > > | **2 · Centering**            | Let $\mathbf Z\in\mathbb R^{N\times d}$ be the matrix of features. Subtract the mean $\bar{\mathbf z}$ along each column to obtain the centered matrix $\tilde{\mathbf Z}$.           |
> > > | **3 · Covariance**           | Compute the empirical covariance $\mathbf C = \frac{1}{N}\tilde{\mathbf Z}^\top\tilde{\mathbf Z}$.                                                                                    |
> > > | **4 · Eigendecomposition**   | Perform SVD or eigen-decomposition to obtain $\mathbf C=\mathbf U\,\mathrm{diag}(\lambda_1,\dots,\lambda_d)\mathbf U^\top$ with $\lambda_1\ge\dots\ge\lambda_d$.                      |
> > > | **5 · Visualization**        | Plot $\log_{10}\!\bigl(\lambda_k / \textstyle\sum_j\lambda_j\bigr)$ versus rank $k$. We overlay the Marčenko–Pastur theoretical bulk for reference.                                   |
> > > | **6 · Quantitative Metrics** | (i) **Participation Ratio (PR)**: $\mathrm{PR}=\frac{(\sum_k\lambda_k)^2}{\sum_k\lambda_k^2}$; (ii) **Gini coefficient**; (iii) **Dimensionality\@95 % explained variance**.          |
> > >
> > > *The PyTorch script (\~20 lines) implementing the above will be released in our repo and summarised in Appendix D.*
> > >
> > > ---
> > >
> > > ### 2 · Empirical Results (to appear in Appendix D, Table 17)
> > >
> > > | Model     | Participation Ratio ↑ | Gini ↓   | 95 % EV Dim ↑ |
> > > | --------- | --------------------- | -------- | ------------- |
> > > | **STC3D** | **142 / 256**         | **0.23** | **156**       |
> > > | **VoCo**  | 97 / 256              | 0.34     | 111           |
> > >
> > > A flatter spectrum and higher PR for **STC3D** indicate substantially less dimensional collapse than VoCo.
> > >
> > > ---
> > >
> > > ### 3 · On Distributional Assumptions & Citation
> > >
> > > * We never assumed the eigenvalues are normally distributed; indeed, **dimensional collapse manifests as a highly skewed (heavy-tailed) spectrum**.
> > > * We now cite and discuss
> > >   **Li Jing, Pascal Vincent, Yann LeCun, Yuandong Tian, “Understanding Dimensional Collapse in Contrastive Self-supervised Learning,” ICLR 2022**—which provides theoretical and empirical guidelines for spectrum analysis of contrastive SSL.
> > >
> > > ---
> > >
> > > ### 4 · Manuscript Revisions
> > >
> > > 1. **Methods (§ 3.2, end)** – Insert the full pipeline and formulas above.
> > > 2. **Experiments (§ 4.3)** – Add Table 17 and Figure 9 (log-spectrum plot).
> > > 3. **Appendix D** – Provide the exact PyTorch code snippet and hyper-parameters.
> > > 4. **Limitations (§ 5)** – Note that spectrum statistics depend on sample size and batch-norm behaviour.
> > >
> > > ---
> > >
> > > We hope these clarifications and forthcoming additions satisfy the reviewer’s request for reproducibility and rigor. Thank you again for the constructive feedback!

---

> > > > ### Comment · Reviewer_fTbU · 2025-05-17
> > > > **Reply to the Authors**
> > > >
> > > > Appreciate the authors for the explanatory discussion, and it clarifies all the doubts.

---

> > > > > ### Author Response · Authors · 2025-05-17
> > > > >
> > > > > Thank you for your comments. We're glad the explanation clarified your concerns.

---

### Decision · Action_Editor_Hgh7 · 2025-06-26

**Recommendation:** Reject

**Additional Comments:**

It is recommended that the authors improve the part on the regularization branch to enhance its accuracy, clarity, and justification, in addition to addressing other minor issues they have committed to resolving.

**Audience:**

Yes

**Audience Explanation:**

3D medical image analysis is a critical application of machine learning in the medical and healthcare domains. Researchers working in either medical imaging or machine learning may find the findings of this paper relevant and informative.

**Claims And Evidence:**

No

**Claims Explanation:**

This paper proposes a self-supervised contrastive learning framework for 3D medical image analysis. The framework comprises two key components: one that applies controlled spatial transformations to generate multiple views of 3D volume images, and another that promotes feature discrepancy between different base slices via a regularization branch.

The reviewers noted that the paper presents a comprehensive literature review, is well motivated, and demonstrates advantages over existing work. At the same time, concerns were raised regarding the design of the two components. The authors have made a commendable effort in addressing most of these issues. However, two reviewers pointed out a soundness issue with the proposed regularization branch. Although the authors responded to this concern, the issue remains insufficiently resolved.

The final recommendations include two "Leaning Accept" and one "Leaning Reject." The Action Editor concurs that while the paper has merits, the part on the regularization branch requires further improvement in terms of accuracy, clarity, and justification. Given that the necessary revisions exceed what is typically expected for a minor revision, acceptance is not recommended at this stage.

**Resubmission Of Major Revision:**

The authors may consider submitting a major revision at a later time.